# SG×P : A Sorghum Genotype × Phenotype Prediction Dataset and Benchmark

**Zeyu Zhang, Robert Pless**
George Washington University
`zeyu|pless@gwu.edu`

**Nadia Shakoor**
Donald Danforth Plant Science Center
`NShakoor@danforthcenter.org`

**Austin Carnahan, Abby Stylianou**
Saint Louis University
`austin.carnahan|abby.stylianou@slu.edu`

## Abstract

Large scale field-phenotyping approaches have the potential to solve important questions about the relationship of plant genotype to plant phenotype. Computational approaches to measuring the phenotype (the observable plant features) are required to address the problem at a large scale, but machine learning approaches to extract phenotypes from sensor data struggle without access to (a) sufficiently large, organized multi-sensor datasets, (b) field trials that have a large scale and significant number of genotypes, (c) full genetic sequencing of those phenotypes, and (d) datasets sufficiently organized so that algorithm centered researchers can directly address the real biological problems. Here, we present SG×P , a novel benchmark dataset from a large-scale field trial consisting of the complete genotype of over 300 sorghum varieties, and time sequences of imagery from several field plots growing each variety, taken with RGB and laser 3D scanner imaging. To lower the barrier to entry and facilitate further developments, we provide a set of well organized, multi-sensor imagery and corresponding genomic data. We implement baseline deep learning based phenotyping approaches to create baseline results for individual sensors and multi-sensor fusion for detecting genetic mutations with known impacts. We also provide and support an open-ended challenge by identifying thousands of genetic mutations whose phenotypic impacts are currently unknown. A web interface for machine learning researchers and practitioners to share approaches, visualizations and hypotheses supports engagement with plant biologists to further the understanding of the sorghum genotype×phenotype relationship. The full dataset, leaderboard (including baseline results) and discussion forums can be found at `http://sorghumsnpbenchmark.com`.

## 1 Introduction

A plant's phenome is defined by its physical and biochemical characteristics, and is the result of the interaction of its genome and its environment. Finding large scale associations between genetic data and measured phenotypes in realistic conditions has the potential to transform real world agricultural systems, by enabling knowledge-driven breeding to improve crop output or performance. This is especially important as there is an ever increasing need for crops that are, for example, more capable of feeding the world, more drought tolerant or otherwise resistant to climate change, or less in need of energy intensive inputs like synthetic fertilizers. In this work we promote research in this area by creating benchmark datasets with large scale, multimodal imagery and genetic data from hundreds of sorghum crop variants grown in real-world field conditions. Sorghum is immensely important for both food and energy purposes, as it is used extensively as a grain crop and as a source of biofuel, and has proven to be productive in a variety of environments and resilient to an ever changing climate.

37th Conference on Neural Information Processing Systems (NeurIPS 2023) Track on Datasets and Benchmarks.

Better understanding the relationship between sorghum genotypes and their expressed phenotypes has the potential to significantly impact the use of sorghum in food and energy production systems.

Plant breeding pipelines classically measure plant features that are thought to be important, and then breed new varieties from those that had desirable features. Features may include appearance, structure and biochemistry (e.g. how much starch is in the stem). Here, our focus is on physical characteristics that are observable from visual imaging modalities. These characteristics, or phenotypes, include things like leaf shape, leaf counts, stalk height, width, and tillering (the count of how many stalks one plant produces). Tools like PlantCV [17] define explicit approaches to measuring these phenotypes under specific, typically indoor, imaging conditions. Learning based approaches to plant phenotyping are improving over time, and new tools like Segment Anything [26], suggest that deep learning advances can also improve classic computer vision approaches to estimating these features.

When there are very large datasets, modern deep learning offers an approach for *automatically* discovering features that best support learning tasks, rather than defining a priori the most relevant features to then measure. Dubbed "Latent Space Phenotyping" [18, 54], this data driven approach uses deep learning to compute features optimized to help solve a learning problem, and then explores the use of that learned feature space as a digital phenotype. The output feature vector produced by such a model is a by-product of the deep learning network finding feature abstractions to correctly predict data labels. If the prediction task is complex enough, the learned features will, ideally, reflect semantically interesting phenotypes that are in the data, and perhaps even phenotypes, or combinations of phenotypes, that are not already known to be relevant.

Our contribution is the Sorghum Genotype × Phenotype (SG×P) Benchmark, a large scale, multimodal benchmark to support discovery of genotype×phenotype relationships. It consists of:

- carefully organized RGB and 3D imagery from a large field site growing over 300 different sorghum varieties and genetic data derived from full genome sequencing of those varieties,
- benchmarks focused on the classification of images based on a small number of single nucleotide polymorphisms (SNPs) known to have phenotypic impact, and over 2,700 SNPs which the impacts are currently unknown,
- implementation and evaluation of baseline deep learning based latent phenotyping, to support comparison with alternative approaches, and
- a web platform supporting comparison of approaches with algorithm performance leaderboards, and discussion forums for interaction between machine learning practitioners and sorghum-focused plant biologists in understanding the SNPs with unknown expression.

The images and dataset files that comprise this benchmark, along with the leaderboards and discussion forums, can be found at `http://sorghumsnpbenchmark.com`.

## 2 Background

### 2.1 Large Scale Datasets and Deep Learning for Plant Phenotyping

Plant and agricultural datasets have been adopted both for general machine learning development – such as the Iris Dataset, a flower sub-species classification dataset, initially introduced by [20] in 1936 and still used today for prototyping and testing simple machine learning algorithms – as well as for more significant plant biology studies that utilize machine learning. In the last decade, deep learning has yielded huge improvements on a variety of computational tasks, including many in plant science. To achieve this performance, deep learning based approaches require very large datasets. Early machine learning focused plant datasets explored variation in the shape and appearance of leaves from large numbers of trees, including the Flavia dataset [64], the MalayaKew Plant Leaf dataset [30] and LeafSnap [28]. In recent years, the scale of the datasets has grown. For example, the Herbarium 2022 Challenge, focused on the fine grained classification of plant species, is comprised of over 1.05 million images of 15,501 vascular plants [33].

While species identification is a common deep learning task in plant science, there are a large number of datasets and papers that focus on more specific plant phenotyping tasks – measuring or characterizing visual features of plants that have been defined by biologists as being interesting or relevant. Most work in this space focuses on performing a specific task – such as fruit or flower detection and segmentation [4, 27, 31, 44, 58, 15, 22], cultivar and species identification [3, 5, 24, 31, 42, 43, 57], plant disease classification [61, 19, 38, 52, 61], leaf counting or shape analysis [1, 16, 21, 35, 55, 60], yield prediction [12, 34, 41, 59], and stress detection [2, 10, 11, 36], among other tasks. There has also been work to develop broader datasets and benchmarks that include multiple phenotyping tasks, such as [37], a large scale dataset of Arabidopsis and tobacco imagery with significant annotations, focused on eight different phenotyping tasks.

In general, this prior work focuses on developing models and algorithms to extract phenotypes that are defined by biologists, as opposed to discovering novel phenotypes or incorporating plant genetics in any way. Latent Space Phenotyping [54] is notable in that it uses deep learning to discover and represent *unknown* plant phenotypes. Those characteristics of a plant are a complex function of its genetics and environmental factors, factors not often incorporated into these deep learning models. Some example work that does include these factors include [32, 45], which develops approaches to predict traits from genetic information, [53] which generates 3D reconstructions to identify leaf-angle related loci in the sorghum genome, or, most related to this benchmark, [62] which uses deep learning to predict the relative functional importance of specific genetic markers and mutations in plants.

In the SG×P Benchmark presented in this paper, we seek to provide an unprecedented scale of curated visual and genomic data and tasks focused specifically on discovering unknown relationships between the sorghum genotype and its expressed phenotypes.

## 2.2 TERRA-REF

The image and genomic data in the SG×P Benchmark was collected as part of the TERRA-REF project [9, 29]. The TERRA-REF gantry system shown in Figure 1 can monitor over an acre of crops with imaging sensors including stereo-RGB, thermal, short- and long-wave hyper-spectral cameras, and laser 3D-scanner sensors. The TERRA-REF data includes imagery from several seasons where plants in the sorghum Bioenergy Association Panel (BAP) [8]) were grown. The BAP is a collection of 390 sorghum lines which have shown potential for bio-energy usage, and all 390 lines were fully sequenced.

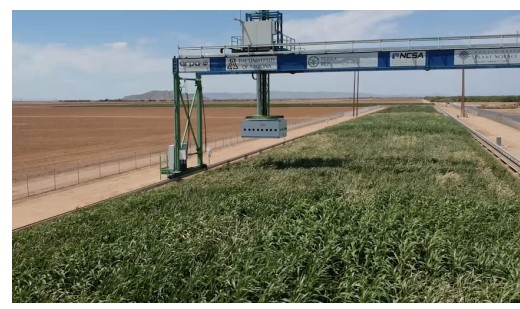

Figure 1: The TERRA-REF Field and Gantry-based Field Scanner in Maricopa, Arizona.

The SG×P Benchmark includes imagery and genetic information curated from the original, full TERRA-REF dataset to be broadly accessible by the machine learning community. It is organized to (a) simplify challenges that arose in capturing data across multiple sensors from differing capture times, sensor failures, and weather conditions, and (b) construct a dataset and evaluation protocol to make the genotype×phenotype comparison accessible to those without a strong biological background. The original TERRA-REF dataset and the derived datasets used in the SG×P Benchmark are released under a CC0 1.0 Universal Public Domain Dedication license.

## 2.3 Sorghum and the Genotype×Phenotype Relationship

Sorghum is a genus of grass plants, and a critical crop globally for human and livestock consumption, and for bioenergy. Sorghum's drought tolerance and ability to grow in harsh environments make it a valuable crop for food security in regions with unreliable rainfall, or regions being impacted by climate change, and currently provides vital nutrition in many developing countries. Furthermore, sorghum has a high potential for biofuel production – its high biomass yield and ability to thrive in varying environmental conditions make it a sustainable alternative to fossil fuels. A deeper understanding of the genotype × phenotype relationship – the relationship between the plant's genetics and its actual expressed traits – is crucial for breeding improved sorghum crops for food and energy purposes. The complex interplay between genes and expressed phenotypic traits in sorghum can impact various characteristics such as yield, drought tolerance, nutrient content, and biomass accumulation.

Identifying the impact of specific genes on plants and their interaction with the environment is a crucial area of research in plant biology [6, 7, 13, 39, 46], and the SG×P dataset and benchmark seeks to advance the understanding of the sorghum genotype×phenotype relationship, and the ways that machine learning and computer vision can be used to uncover these relationships.

## 3 SG×P Benchmark

A high level overview of the plant and sorghum genetics helps to understand the data and labels in this benchmark. Sorghum has 10 chromosomes, each of which contains genetic instructions in the form of DNA – these genetic instructions drive both the form and function of the plant. Sorghum is a diploid species, meaning that it has two copies of each of these 10 chromosomes. The DNA that comprises these chromosomes is composed of nucleotides (either A,C,T,G), which form sequences that direct the plant to produce specific proteins. Variations at specific locations in these sequences, can be defined by a change of a single nucleotide (e.g. an A is changed to a G). These variations are

| Chromosome | Gene | Position | Known Controlled Phenotype |
|---|---|---|---|
| 1 | 001G269200 | 51588525 | |
| 1 | 001G269200 | 51588838 | Wax composition [56]) |
| 1 | 001G269200 | 51589143 | |
| 1 | 001G269200 | 51589435 | |
| 6 | 006G067700 | 42805319 | Plant height and structure, stem length and internode length [67, 25]) |
| 6 | 006G067700 | 42804037 | |
| 6 | 006G147400 | 50898459 | |
| 6 | 006G147400 | 50898536 | |
| 6 | 006G147400 | 50898315 | Plant height and structure, and sugar composition [66]) |
| 6 | 006G147400 | 50898231 | |
| 6 | 006G147400 | 50898523 | |
| 6 | 006G147400 | 50898525 | |
| 6 | 006G057866 | 40312463 | Flowering time and maturity [14, 63, 40]) |
| 6 | 006G004400 | 2697734 | |
| 9 | 009G229800 | 57040680 | Pigmentation and tannin production [65]) |

Table 1: Details about the SNPs in the dataset with known phenotypic expression.

called single nucleotide polymorphisms (SNPs), and they are important as they can alter the proteins produced by the plant, affecting its structure and performance.

Because Sorghum is diploid, a particular line of sorghum may have a variation on one or both copies of its chromosome. If the variation is on only one copy, that line of sorghum is termed heterozygous for that SNP; if the variation is on both copies, it is called homozygous. We only consider homozygous cases to simplify the machine learning task and the biological interpretation. Additionally, while in principle there could be different variations at the same location, in our case each location is unique (so there is no case where at one SNP location an A is changed to a C in some lines, and the same A is changed to a G in other lines).

## 3.1 Tasks

The overarching task in this benchmark is a binary classification task: to predict whether a plant has the reference or alternate version of a SNP based on an image or images of that plant from one or more imaging modalities. The benchmark includes datasets with RGB and 3D images for 15 SNPs with known genotype×phenotype relationships, and 2,717 SNPs with unknown genotype×phenotype relationships (the specifics of the data are detailed in Section 3.2).

Details about the 15 SNPs with known phenotypic expression can be found in Table 1 – these SNPs are known to control phenotypes such as leaf wax composition, plant height and structure, pigmentation, and flowering time. In prior work [50, 68], we showed that simple classification models trained on RGB images can accurately predict whether an image shows a plant with a reference or alternate version of many of these SNPs. Additionally, heatmap-style explanatory visualizations from these models often focused on the same traits that the SNPs were known to control. An example of this is shown in Figure 2 (right), which shows an image of a sorghum plant that has the reference version of a SNP that controls leaf wax composition, and the corresponding visualization for why a neural network trained on the binary classification task predicted the reference label, highlighting that the network learned to focus on the waxy build-up on the plant.

We encourage machine learning researchers engaging with this benchmark to first develop and evaluate approaches on these markers with known phenotypic expression – all of these markers have visual traits that are detectable and machine learning approaches to perform binary classification task on these datasets can achieve significantly above chance performance. This makes these SNPs useful for getting comfortable with the data in the benchmark and checking the validity of approaches.

These 15 SNPs, however, are perhaps less exciting to consider than the 2,717 SNPs in the SG×P Benchmark that have *unknown* phenotypic expression. While many SNPs are "synonmyous" – meaning that the nucleotide change does not result in an amino acid change in the protein sequence, and so no change is made to the structure, function or interaction of the encoded protein with other molecules – these 2,717 SNPs were identified as potential "high impact" polymorphisms. When a SNP is referred to as potentially high impact, it means that the nucleotide change in that SNP has the potential to significantly alter the protein coding sequence or its function. Many of these changes may not result in obvious changes to the phenotype – but some certainly will. Finding those SNPs and uncovering their phenotypic expression is exactly the goal of this benchmark, and will be key to better understanding the genotype×phenotype relationship in sorghum and developing new lines of sorghum that have desirable properties.

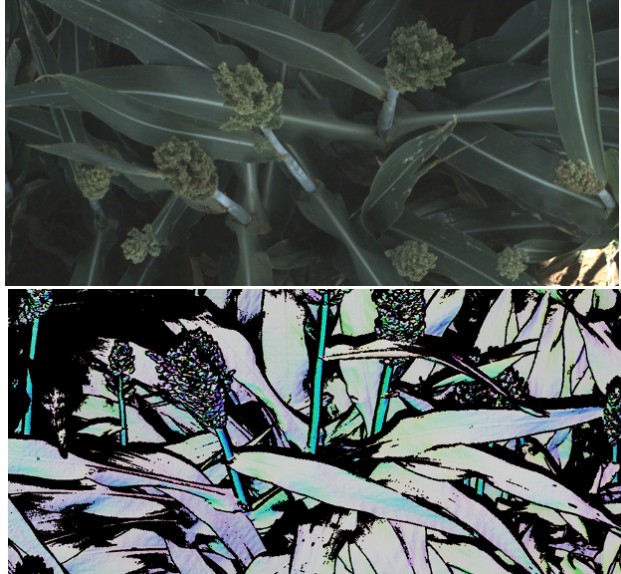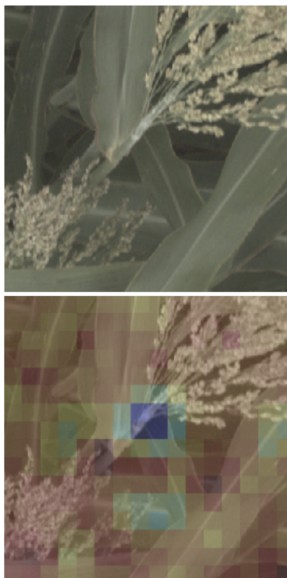

Figure 2: (left-top) Example RGB imagery from the dataset. (left-bottom) A false color 3D line scanner image from approximately the same location and date as the RGB image. The line scanner data is shown with false coloring that highlights surface normal. (right) An image of a sorghum plant that has the reference version of a SNP that controls leaf wax composition, and the class activation map showing the model trained on this task learned to focus on wax buildup.

### 3.2 Data

The entire SG×P Benchmark consists of 543,574 RGB images and 536,153 false color 3D line scanner images from the 2017 TERRA-REF growing season. There are 105 days with data captured over the course of the 140 day growing season; some days did not have any data capture. Each sorghum cultivar in the SG×P Benchmark was grown in two spatially separated plots within the field. We provide images from both plots in our datasets, organized by cultivar.

To construct a simple image dataset, RGB images that cross plot boundaries are split into multiple images, so that every image contains pixels of plants from only a single plot. This guarantees that all images in our benchmark only show images of a single cultivar. The 3D line scanner output is also cropped to plot boundaries. We convert the raw line scanner output from a three dimensional voxel representation to a two dimensional false color image, where the hue encodes the surface normal at each pixel location, and the value encodes distance from the camera (completely black pixels are locations that were occluded). An example of this false color representation of the 3D line scanner output can be seen in Figure 2 (bottom). For convenience, we will refer to these false color images as the 3D images in the remainder of the paper. Researchers interested in using other representations of the 3D data are welcome to download it from the original TERRA-REF data [29], but should include information about their processing and data format when submitting results to the leaderboard.

We use these images across all of the SNP prediction datasets. There are 15 datasets focused on SNPs with known phenotypic expression and 2,717 SNPs with unknown phenotypic expression. For every one of these SNPs, we provide six datasets: RGB training and testing datasets, 3D training and testing datasets, and multimodal training and testing datasets that contain 3D-RGB pairs. The reason we consider a multimodal benchmark in addition to benchmarks for each separate modality, is that we believe there may be interesting and complex phenotypes that are observable using multiple modalities that may not be observable using only one modality (e.g., a variation in a particular SNP might produce taller *and* greener leaves).

For every image including in any dataset in the SG×P Benchmark, we also provide additional metadata: the sorghum cultivar and subpopulation shown in the image, the TERRA-REF plot from which the image was captured, and the timestamp that the image was captured. Additional information about this metadata can be found in the Appendix. Benchmark participants are encouraged to explore ways to incorporate this metadata information in their approaches.

In order to generate training and testing splits for each SNP, we find whether there are more reference or alternate cultivars. We take a randomly selected third of the cultivars from whichever label has fewer cultivars and put their images in the SNP-specific test set. We then complete the test set with images from an equal number of randomly selected cultivars of the other label. In order for a SNP to be included in the SG×P Benchmark, we require that there are at least 20 reference and 20 alternate cultivars. This means that there is a minimum of 6 reference and 6 alternate cultivars in each test set. We then put all other cultivars in the training set, maximizing the amount of data available for training, and guaranteeing that there is no overlap in cultivars between the training and testing sets.

Figure 3 shows distributions of images and labels for different subsets of the SG×P dataset. Notably, Figure 3 (c) shows the training data distribution of reference minus alternate labels per SNP, normalized by the total number of images in the dataset. Points that are farther to the left have more alternate images than reference images, and points that have more reference images than alternate images. This plot shows that the training datasets often have significant label imbalance, typically with more reference images than alternate images. The test sets are all balanced to have an equal number of reference and alternate images (guaranteeing that the average random model achieves 50% accuracy on the binary classification task). We make sure there is no overlap in the cultivars included in the training and testing datasets. This ensures that models cannot overfit on trivial, non-biologically relevant features (such as unique patterns in the dirt or in the lighting conditions). Instead, models that generalize from the cultivars in the training datasets to the entirely different cultivars in the testing datasets are learning biologically relevant visual features.

Figure 2 shows an example pair of multimodal images. Although reasonably well aligned, the alignment between sensors is not perfect because the RGB camera and the 3D line scanner camera are physically in different parts of the gantry sensor box, and because the RGB images and 3D line scanner images are typically captured at different times of day. This lack of spatial and temporal alignment is a challenge for multi-sensor fusion approaches, and different approaches are possible.

Our approach to creating simple multimodal training and testing datasets is to provide pre-computed 3D-RGB data pairs. To create the dataset we took each of the 3D images from the SNP-specific datasets and selected a random RGB image from the closest date in time and the same plot, to create 3D-RGB pairs. We exclude any 3D images that do not have a nearby RGB image from within 10 days. In this construction, some RGB images that are included in the SNP-specific RGB datasets may not be used, some 3D images may not have a valid RGB match, and some RGB images may be the match for multiple 3D images. More details on the multimodal dataset construction are in the Appendix. While we believe these RGB-3D pairs are a useful data structure for initial exploration of multimodal sensor fusion for genotype×phenotype discovery, we encourage practitioners utilizing this benchmark to consider other approaches that may be more complex.

## 4 Baseline Approach

We provide baseline deep learning models and results for all SNPs in the SG×P Benchmark. Prior to training task specific baseline models for each of the SNPs, we pre-train a ResNet-50 model [23] for RGB and 3D imagery, to learn useful, generalizable sorghum features. To do this pre-training, we leverage additional publicly available data from the TERRA-REF project – 110,638 RGB images and 141,113 3D scanner images from 188 cultiavrs grown in 2019, processed in the same way as the data in this benchmark. The 188 cultivars that are part of this 2019 dataset have no overlap with the ones in the SG×P Benchmark, but have significant phenotypic variation, meaning models trained on this data learn useful visual features for tasks relating to sorghum.

The pre-trained ResNet-50 models are trained to classify which of 375 field plots an image comes from using cross-entropy loss. We train one model with RGB imagery and one with 3D imagery. The models have a 2048-dimensional global average pooling layer, followed by a 128-dimensional fully connected layer, and then the plot level classification output. During this pre-training, images (RGB or 3D) are resized to 512x512 pixels and randomly flipped (horizontally and vertically). We use PyTorch, training for 25 epochs with the SGD optimizer with an initial learning rate of 0.01, and a learning rate decay of 0.1 every 10 epochs. The pre-trained model takes roughly one day to train on a single 16GB P100 Nvidia GPU with a batch size of 30.

After pre-training, we chop off the final plot classification layer. For each SNP prediction task in the SG×P Benchmark, we then take this pre-trained model and train a single fully connected layer with two outputs, corresponding to the Reference and Alternate labels, using the SG×P Benchmark training datasets (because the pre-trained model was trained on data from an entirely different growing season, there is no risk of data leakage or overfitting). For the multi–modal tasks, we first concatenate the 128-dimensional output features extracted by each of the RGB and 3D pre-trained models and

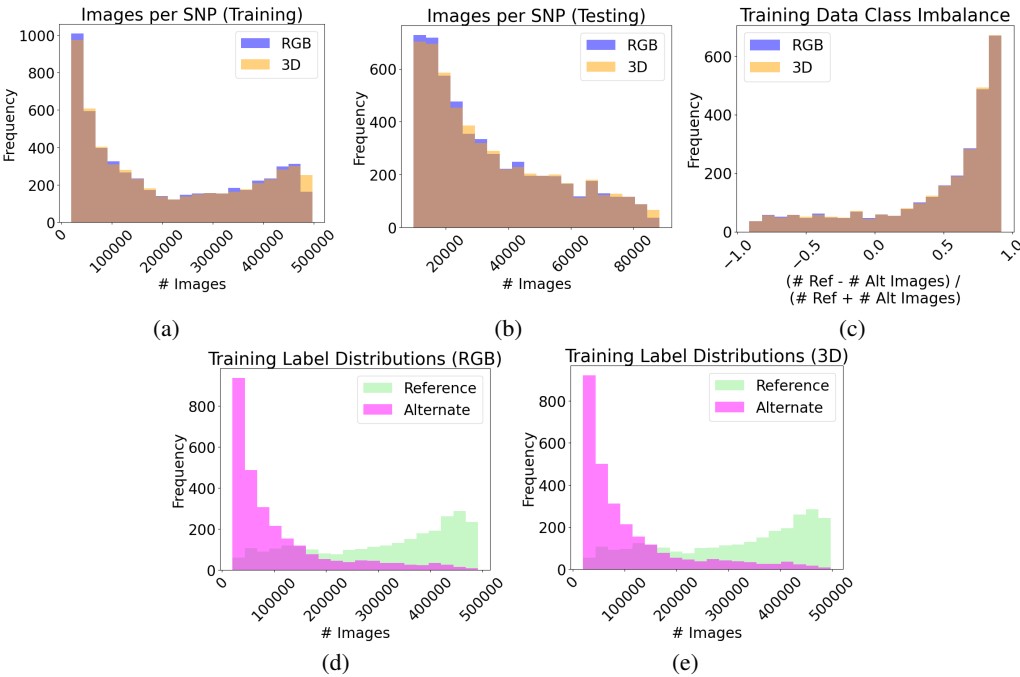

Figure 3: These plots show distributions of number of images by dataset, by sensor and by label. Figure (c) in particular conveys the class imbalance in the training data, showing the distribution of reference images minus alternate images in the training data (normalized by the total number of images for that dataset). Datasets with more reference images are farther to the right, and tasks with more alternate images are farther to the left.

then train a single fully connected layer on top of those concatenated features. In order to facilitate fast training of these SNP-specific models, we first extract the 128-D output features for every image in the dataset, and then train only the final fully connected layer for each SNP classification task. We use a batch size of 256 when training the SNP-specific models. Due to significant class imbalance in many of the datasets, we use a balanced sampler to guarantee a roughly equal number of Reference and Alternate examples in each training batch. We discuss the results of this baseline in Section 5, and provide links to the data used for pre-training, as well as our pre-trained model and training code, at the benchmark website (`http://sorghumsnpbenchmark.com`).

## 5 Evaluation

For every SNP in the SG×P Benchmark, we consider two different evaluation paradigms for each of the imaging modalities: one that is per-image, and one that is per-cultivar. The evaluation metric is binary classification accuracy and per-image evaluation is straight-forward – for every image in the test dataset, is the predicted SNP label correct or not? This is averaged over the entire test set (every test set in the SG×P Benchmark has an equal number of Reference and Example images, meaning that the average random model would achieve 50% accuracy on each task). We report on the results of the baseline approach described in Section 4 for the SNPs with known phenotypic expression in the middle three columns of Table 2. In all cases, the baseline models achieve significantly above chance performance, with the multimodal approach often outperforming the single-modality approaches.

This per-image evaluation is valuable for three reasons. First, many baseline ML classification approaches are designed to ingest a single image or image pair and produce a classification. Second, many variations of explainable image classification [47, 48, 49, 51, 69] operate on single images at a time to highlight visual features that affected the classification result. This approach to explainability can be useful when trying to understand genotype×phenotype interactions.

The per-image evaluation, however, is not necessarily always the most appropriate. Some genetic variations may only impact the plant during a small portion of the growing season – for example, causing changes to the shape, size or timing of flowers that the plant produces. These sorts of variations would not be meaningfully correlated with visual features in images from other parts of the growing season. The goal of the SG×P Benchmark is to uncover meaningful and as-yet unknown

| | SNP | | By Image | | | By Cultivar | | |
|---|---|---|---|---|---|---|---|---|
| Chromosome | Gene | Position | RGB Only | 3D Only | RGB+3D | RGB Only | 3D Only | RGB+3D |
| 1 | 001G269200 | 51588525 | 0.615 | 0.569 | 0.640 | 0.640 | 0.700 | 0.700 |
| 1 | 001G269200 | 51588838 | 0.601 | 0.564 | 0.582 | 0.652 | 0.696 | 0.609 |
| 1 | 001G269200 | 51589143 | 0.622 | 0.597 | 0.637 | 0.662 | 0.662 | 0.676 |
| 1 | 001G269200 | 51589435 | 0.640 | 0.606 | 0.665 | 0.781 | 0.750 | 0.812 |
| 6 | 006G067700 | 42805319 | 0.604 | 0.584 | 0.630 | 0.654 | 0.654 | 0.744 |
| 6 | 006G067700 | 42804037 | 0.597 | 0.571 | 0.620 | 0.653 | 0.681 | 0.667 |
| 6 | 006G147400 | 50898459 | 0.539 | 0.536 | 0.578 | 0.553 | 0.605 | 0.684 |
| 6 | 006G147400 | 50898536 | 0.505 | 0.569 | 0.578 | 0.500 | 0.684 | 0.658 |
| 6 | 006G147400 | 50898315 | 0.527 | 0.584 | 0.600 | 0.553 | 0.658 | 0.711 |
| 6 | 006G147400 | 50898231 | 0.617 | 0.578 | 0.650 | 0.735 | 0.618 | 0.765 |
| 6 | 006G147400 | 50898523 | 0.578 | 0.568 | 0.626 | 0.553 | 0.579 | 0.632 |
| 6 | 006G147400 | 50898525 | 0.625 | 0.576 | 0.650 | 0.763 | 0.658 | 0.842 |
| 6 | 006G057866 | 40312463 | 0.670 | 0.543 | 0.659 | 0.850 | 0.650 | 0.850 |
| 6 | 006G004400 | 2697734 | 0.624 | 0.630 | 0.709 | 0.763 | 0.816 | 0.842 |
| 9 | 009G229800 | 57040680 | 0.587 | 0.590 | 0.636 | 0.611 | 0.704 | 0.648 |

Table 2: Binary classification performance of the baseline approach on the SNPs with known genotype×phenotype relationships. Performance is reported for each sensor separately, and when using sensor fusion, for both individual images and when aggregated by cultivar.

genotype×phenotype relationships, including those that may not be observable in every image in the benchmark. To support this goal, we provide a more flexible, per-cultivar evaluation.

In the per-cultivar evaluation, researchers provide a single prediction for each of the cultivars in the test set (as opposed to each image). As in the per-image evaluation, the number of cultivars that have the Reference versus Alternate labels is the same, so the average random model would achieve 50% accuracy. We note that the number of cultivars per label for some SNPs can be relatively small – e.g. about 10 – meaning that we expect there to be more noise in the average accuracy of models on this evaluation. In the results shown in the last three columns of Table 2, we used the simplest approach to per-cultivar evaluation: we decided whether a cultivar is Reference or Alternate for a particular SNP by taking the mode of the predictions from all of the images that belong that cultivar.

Researchers have flexibility in how they produce the per-cultivar prediction. They could, for example, only consider images from a particular portion of the growing season, or weight images from different parts of the growing season differently. They could develop model architectures that consider multiple images simultaneously. In general, the per-cultivar evaluation provides researchers with the ability to develop models that explain complex genotype×phenotype relationships in sorghum.

One way that the SG×P Benchmark varies from traditional benchmarks is that it contains a significant number of tasks where it is unknown if above-chance performance is even possible. While we have structured the SG×P Benchmark to include SNPs that are already well understood for proof of concept and validation, in addition to the SNPs with unknown genotype×phenotype interations that are potentially "high impact", in reality, many of these unknown SNPs will have minimal impact. While this uncertainty makes our benchmark unique in machine learning datasets and competitions, it is not unique in science – the goal of this benchmark is discovery.

## 6 Web Page, Leaderboards & Discussion Forums

To support that discovery, we provide a web interface that has not only traditional leaderboards where researchers can compare their numerical performance to others, but also discussion forums where they can interact with plant biologists, sharing about their approaches, what they think their models are learning, and participate in back and forth dialogues to drive this discovery of unknown genotype×phenotype relationships in sorghum. The website can be found at `http://sorghumsnpbenchmark.com`. Screenshots of this web interface can be found in Figures 4 and 5.

Figure 4 (left) shows the main page for the SG×P Benchmark. This page includes leaderboards for each of the SNPs in the Benchmark, with their current top accuracy by image and by cultivar, as well as the attribution for those leading results. Users can switch between leaderboards for each of the different imaging modalities. By clicking on the SNP name on these leaderboards, users are taken to a detail page for that SNP (seen in Figure 4 (right)). This page includes metadata for the SNP (its chromosome, gene and position on the chromosome), details about each of the training and testing datasets including links to download the data, SNP-specific leaderboards for each of the imaging modalities, and links to any Discussion Board posts that are about that SNP.

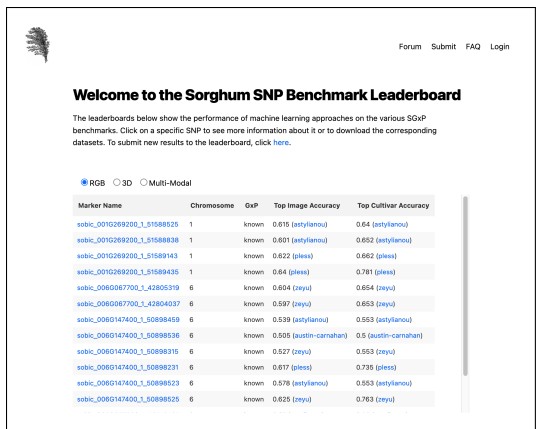
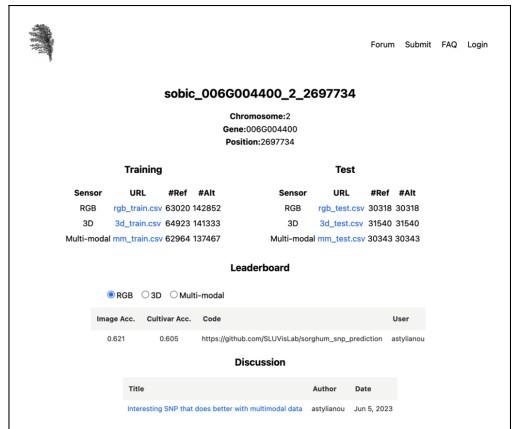

Figure 4: (left) The overall leaderboard for the SG×P Benchmark, showing the current top results for different SNPs. (right) The detail page for one of the SNPs, including metadata, dataset information, the current SNP-specific leaderboards, and discussion board posts about the SNP.

In Figure 5, we show one such example discussion for the SNP highlighted in Figure 4 (right). This SNP was interesting because the baseline model achieved well above chance accuracy in all modalities, but also saw a significant improvement from using the multimodal data. We show an example discussion, where we posted about this interesting behavior and one of the biologists on our team replied with information from the sorghum literature about the SNP and a follow up asking about visual features that the machine learning model was focusing on. This back-and-forth engagement is precisely the goal of the SG×P Benchmark – to drive the discovery of genotype×phenotype relationships that can lead to new understanding about sorghum and direct development of new and improved sorghum crops.

# 7 Limitations

There are several possible limitations of this work. The focus on sorghum may limit the generalizability of the findings. While we believe the structure of the machine learning approaches that on this dataset would generalize, other crops may exhibit unique genetic and phenotypic characteristics that require different approaches. The data represents a snapshot of sorghum variants grown in specific field conditions, which may introduce biases and limitations in capturing the full range of genetic and environmental factors that influence sorghum phenotypes. The benchmark is also limited to focusing on above ground traits that are visible in RGB and 3D data – this ignores any traits that are expressed below group or are not visible in these imaging modalities. It also is setup to consider the genetic impact of individual SNPs and does not consider more complex genetic interactions.

The SNPs chosen were ones that have a potential biological impact because they change the amino acid that the DNA encodes. However, that amino acid may not be important to the function of the plant, so many, perhaps most of these SNPs will have no biological impact. This limits the value of these tasks for researchers focussed on demonstrating that generic machine learning algorithms work in this domain. These tasks may be more compelling for researchers that want to be involved in discovering biological function; in this case we provide the discussion forum because an machine learning algorithm that scores well only goes part of the way to discovering the phenotype changed by that SNP, and additional work with visualizations or other explainability approaches may be needed to discover the genotype×phenotype relationship.

# 8 Conclusion

This Sorghum Genotype × Phenotype (SG×P ) Benchmark provides a resource for exploring the relationships between genetic data and phenotypic expression in sorghum. By integrating large-scale, multimodal imagery and genetic information from a large number of sorghum species grown in real-world field conditions, this benchmark enables researchers to uncover significant insights into the use of sorghum in food and energy production systems. The benchmark includes baseline deep learning approaches, allowing for comparison to future algorithms that learn data-driven features relevant to understanding sorghum phenotypes. We create a combined leaderboard and discussion

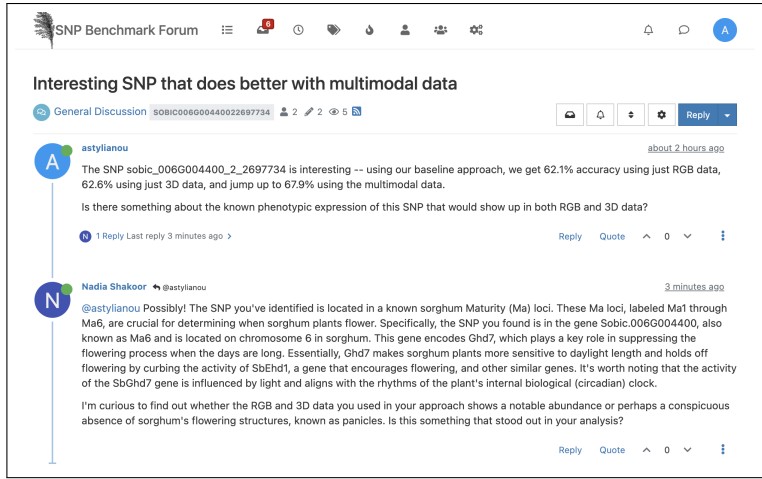

Figure 5: A discussion board post about the SNP shown in Figure 4.

forum to further facilitate collaboration and knowledge-sharing among machine learning practitioners and sorghum-focused plant biologists.

The SG×P Benchmark focuses specifically on sorghum, but we believe that algorithms and methodologies that are effective on this benchmark have the potential to apply to other crops and organisms with similar scales of imagery and genetic data. In building the benchmark, we have worked to make the tasks directly relevant to understanding some part of the relationship between genotypes and phenotypes, to simplify and catalyze interdisciplinary interactions within this research domain. In doing so, we hope that the SG×P Benchmark thus serves as a catalyst to unraveling the intricate connections between genetic information and observable traits in sorghum and beyond, with the ultimate goal of addressing global challenges in food production, ecosystem resilience, and sustainable development.

## Acknowledgments and Disclosure of Funding

This work was funded by the Advanced Research Projects Agency - Energy (ARPA-E) awards DE-AR0000594 and DE-AR0001101, by a seed grant from the Agricultural Genome to Phenome Initiative (AG2PI) (USDA-NIFA awards: 2022-70412-38454, 2021-70412-35233, and 2020-70412-32615), and by Google's Research Scholar Program ("Using Deep Learning to Uncover Unknown genotype×phenotype Relationships in Crops").

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
