# SG×P : A Sorghum Genotype × Phenotype Prediction Dataset and Benchmark

## A    Appendix

### A.1    Additional Metadata

For all RGB and 3D scanner images used in the SG×P Benchmark, we provide additional metadata that Benchmark participants are encouraged to explore using in their approaches. This metadata includes:

- **filepath (str)**: The full filepath for the file in the dataset.

- **subpopulation (int)**: A unique identifier for which subpopulation the plant shown in the image belongs to. A sorghum subpopulation refers to a distinct and identifiable group within the larger sorghum plant species (Sorghum bicolor) that exhibits specific genetic, morphological, or ecological characteristics setting it apart from other groups within the same species. These characteristics can include variations in traits such as plant height, grain color, flowering time, disease resistance, and adaptations to specific environmental conditions, and participants in the SGxP Benchmark may want to incorporate this information in their methodology or analysis.

- **cultivar (str)**: A unique identifier for which line of sorghum the plant shown in the image belongs to. A cultivar, short for "cultivated variety," is a distinct and deliberately bred cultivated plant variety that exhibits specific characteristics such as growth habits, yield potential, disease resistance, and other desirable traits. Cultivars are developed through selective breeding or genetic manipulation to enhance certain features, making them well-suited for particular agricultural or horticultural purposes. To provide a more concise and standardized method of referencing cultivars, a system of alphanumeric codes known as "PI numbers" is used. "PI" stands for "Plant Introduction," and the numbers are assigned sequentially to different plant varieties introduced into a genebank or germplasm collection.

- **plot (int)**: A unique identifier for the TERRA-REF plot the image came from. Each sorghum cultivar represented in the SGxP Benchmark was grown in two spatially separated plots in the TERRA-REF field.

- **timestamp (datetime)**: A precise timestamp for when the image was captured by the TERRA-REF gantry (in UTC-12).

### A.2    Procedure for Generating the Multimodal Dataset

As part of the SG×P Benchmark, we provide a multimodal dataset that consists of pre-constructed pairs of 3D and RGB images. We document the process for constructing these datasets below:

1. For every RGB image, initialize a match_counter to 0 (the number of times this RGB image was used as a match for a 3D scanner image).

2. For every 3D scanner image:
   - (a) Find the RGB images (from the respective single modality dataset) that come from the same plot, were captured within 10 days of the 3D scanner image.
   - (b) Remove from this subset any images whose match_counter is greater than the minimum match_counter value for the entire subset.
   - (c) Select one of the remaining images as a good match.
   - (d) Increment the match_counter for that image.

Some 3D scanner images have no valid RGB image from the same plot within 10 days. These images are excluded from any multimodal datasets. This construction maximizes the number of multimodal image pairs that are included in the dataset (removing only those where the 3D scanner image did not have another image from the same plot within 10 days), while allowing for more

possible repeated RGB images (when there were fewer RGB images captured in the same plot within 10 days).

After creating the list of (3D scanner, RGB) pairs using the above methodology, we create SNP-specific multimodal datasets. To do this, we load the 3D scanner test set for a specific SNP. For every image in that test set, we look up its (3D scanner, RGB) pair. If a valid pair exists, then it is added to the possible multimodal test set for that SNP.

If the dataset is a testing dataset, we then identify any (3D scanner, RGB) pairs where the RGB image is repeated. We select one of the pairs at random to keep, and remove any other pairs, so that the test set does not contain any repeated RGB images. We allow repeated RGB images in the training set to maximize the number of (3D scanner, RGB) pairs available at training time. We finally rebalance the test set using the same procedure as the single modality datasets, guaranteeing that there are an equal number of reference and alternate examples.

Researchers are welcome and encouraged to explore different constructions of multimodal datasets which can be evaluated using the per-cultivar evaluation metric.

## A.3 Performance of Baseline Model Over Time

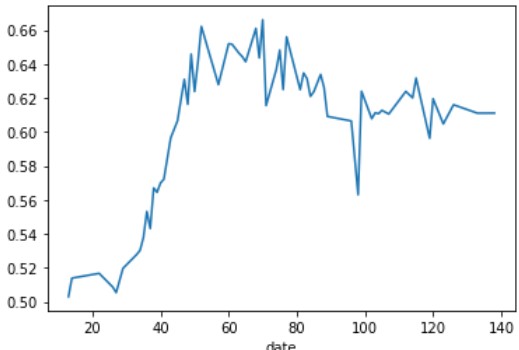

Appendix Figure 1: Average performance of baseline approach on known SNPs per day.

The primary evaluation for the SG×P Benchmark considers the performance of a model over the entire growing season represented in the dataset. We can, however, also evaluate the performance per-day over the course of the growing season. In Figure 1, we show the average per-day performance of the baseline model described in Section 4 for the SNPs with known genotype×phenotype relationships. The x-axis shows the number of days after planting, and the y-axis shows the average performance over all of the SNPs (excluding any days with 5 or fewer SNPs represented). We can see that early in the growing season, when the plants are initially emerging, quite small, and often look similar to each other, performance is, on average, barely above chance (50%). As the plants grow and differentiate, performance rapidly increases. Once the plants start to significantly occlude each other and then begin to die off later in the season, performance starts to decay some. This behavior is not especially surprising, but does suggest opportunities for researchers to explore better models and features for representing early season genotype×phenotype relationships, or for weighting the early season data differently than in our baseline approach.

## A.4 Evaluation of Baseline on Unknown Genetic Markers

In Table 3, we provided the per-image and per-cultivar accuracy of our baseline approach on each of the SNPs in the SG×P Benchmark with known phenotypic expression. For the over 2700 SNPs with unknown phenotypic expression, we report in Appendix Table 1 on statistics of our baseline approach over all of the SNPs with unknown phenotypic expression.

## A.5 Dataset Documentation

In this section, we provide responses to the Datasets for Datasheets framework for dataset documentation [2].

| Statistic | By Image | | | By Cultivar | | |
|---|---|---|---|---|---|---|
| | RGB Only | 3D Only | RGB+3D | RGB Only | 3D Only | RGB+3D |
| mean | 0.590 | 0.570 | 0.619 | 0.667 | 0.666 | 0.710 |
| std | 0.051 | 0.042 | 0.060 | 0.107 | 0.111 | 0.108 |
| $25^{th}$ percentile | 0.555 | 0.541 | 0.577 | 0.600 | 0.591 | 0.640 |
| $50^{th}$ percentile | 0.585 | 0.566 | 0.610 | 0.667 | 0.658 | 0.705 |
| $75^{th}$ percentile | 0.623 | 0.596 | 0.657 | 0.731 | 0.731 | 0.778 |

Appendix Table 1: Performance of the baseline approach on the SNPs with unknown genotype×phenotype relationships. Performance is reported for each sensor separately, and when using sensor fusion, for both individual images and when aggregated by cultivar.

### A.5.1 Motivation

**For what purpose was the dataset created? Was there a specific task in mind? Was there a specific gap that needed to be filled?** The goal of the SG×P Benchmark is to support the development of algorithms that can find genotype×phenotype relationships in bioenergy sorghum, by integrating large-scale, multimodal imagery and genetic information from a large number of sorghum species grown in real-world field conditions. The SG×P Benchmark focuses specifically on sorghum, but we believe that algorithms and methodologies that are effective on this benchmark have the potential to apply to other crops and organisms with similar scales of imagery and genetic data.

**Who created the dataset (e.g., which team, research group) and on behalf of which entity (e.g., company, institution, organization)?** The raw data from which our dataset and benchmark was curated was produced by the TERRA-REF team [1]. Several of the authors of this paper (Zhang, Shakoor, Pless and Stylianou) were members of that team, as students, project management, faculty, and postdocs, respectively, and continue to work on ARPA-E funded projects utilizing the same data and sensors, as students and faculty at George Washington University, the Donald Danforth Plant Science Center, and Saint Louis University.

**Who funded the creation of the dataset? If there is an associated grant, please provide the name of the grantor and the grant name and number.** This work was funded by the Advanced Research Projects Agency - Energy (ARPA-E) awards DE-AR0000594 and DE-AR0001101, and by a gift from the Google Research Scholar Program ("Using Deep Learning to Uncover Unknown genotype×phenotype Relationships in Crops").

### A.5.2 Composition

**What do the instances that comprise the dataset represent (e.g., documents, photos, people, countries)? Are there multiple types of instances (e.g., movies, users, and ratings; people and interactions between them; nodes and edges)?** The dataset consists of RGB and 3D imagery that show overhead views of biomass sorghum. We provide metadata for each image including its cultivar, subpopulation, TERRA-REF plot number, and timestamp. The benchmark consists of over 2700 binary classification tasks, predicting whether an image from the dataset shows plants with a "reference" or "alternate" version of a single nucleotide polymorphisms (SNPs; effectively, whether the image shows a plant with a specific mutation or not).

**How many instances are there in total (of each type, if appropriate)?** The entire SG×P Benchmark dataset consists of 543,574 RGB images and 536,153 false color 3D line scanner images from the 2017 TERRA-REF growing season. The Benchmark consists of 15 binary classification tasks focused on SNPs with known phenotypic expression and 2,717 on SNPs with unknown phenotypic expression. Statistics on the mean and standard deviation of number of images in the training and testing datasets associated with each binary classification task can be found in Table 2.

**Does the dataset contain all possible instances or is it a sample (not necessarily random) of instances from a larger set?** The data included in the SG×P Benchmark was sampled from raw data collected as part of the TERRA-REF project [1]. We include imagery from the 2017 growing season as it had the most complete genetic sequencing, and most complete data coverage over the entire growing season. The SNPs included in the Benchmark were a sample of all of the SNPs from the full genetic sequencing. To be included in the Benchmark, a SNP with unknown phenotypic expression had to (a) be considered "high impact" (potentially changing the protein encoded by the

gene that the SNP is on) and (b) have at least 20 cultivars that are reference and 20 cultivars that are alternate.

**What data does each instance consist of?**   For each image (RGB or 3D), we provide the sorghum cultivar and image timestamp.

**Is there a label or target associated with each instance?**   For each image (RGB or 3D), we provide the label (reference or alternate) for any of the binary SNP classification tasks in the benchmark in which that image is used.

**Is any information missing from individual instances?**   No.

**Are relationships between individual instances made explicit (e.g., users' movie ratings, social network links)? If so, please describe how these relationships are made explicit.**   For each instance (image or 3D sign), we label that instance with the cultivar. Images are related if they are taken of the same cultivar. Other relationships may be inferred based on the genetics of the cultivar, for example, two images may both be viewing the cultivars that share a genetic mutation.

**Are there recommended data splits (e.g., training, development/validation, testing)?**   As part of the SG×P Benchmark, we provide explicit training and testing splits. For every binary SNP classification task in the SG×P Benchmark, we find whether there are more reference or alternate cultivars. We take a randomly selected third of the cultivars from whichever label has fewer cultivars and put their images in the SNP-specific test set. We then complete the test set with images from an equal number of randomly selected cultivars of the other label. Because the criteria for a SNP to be included in the SG×P Benchmark was that there were at least 20 reference and 20 alternate cultivars, this means that there is a minimum of 6 reference and 6 alternate cultivars in each test set. The structuring of the test set means that there is expected random performance of 50%. We then put all other cultivars in the training set, maximizing the amount of data available for training, and guaranteeing that there is no overlap in cultivars between the training and testing sets, requiring models to generalize to achieve high test set performance.

**Are there any errors, sources of noise, or redundancies in the dataset?**   None that we are aware of.

**Is the dataset self-contained, or does it link to or otherwise rely on external resources (e.g., websites, tweets, other datasets)?**   The dataset is self-contained.

**Does the dataset contain data that might be considered confidential (e.g., data that is protected by legal privilege or by doctor– patient confidentiality, data that includes the content of individuals' non-public communications)?**   No.

**Does the dataset contain data that, if viewed directly, might be offensive, insulting, threatening, or might otherwise cause anxiety?**   No.

### A.5.3   Collection Process

**How was the data associated with each instance acquired?**   All data acquired in this project was collected as part of the TERRA-REF project [1].

**What mechanisms or procedures were used to collect the data (e.g., hardware apparatuses or sensors, manual human curation, software programs, software APIs)?**   Imagery data was collected by RGB and 3D line scanner sensors in the TERRA-REF gantry system, and genetic resequencing was performed by Hudson Alpha.

**If the dataset is a sample from a larger set, what was the sampling strategy (e.g., deterministic, probabilistic with specific sampling probabilities)?**   See the answer above on data splits.

**Who was involved in the data collection process (e.g., students, crowdworkers, contractors) and how were they compensated (e.g., how much were crowdworkers paid)?**   Data was collected by funded students, staff and faculty as part of the TERRA-REF project [1].

**Over what timeframe was the data collected?**   The raw data was collected in 2017.

**Were any ethical review processes conducted (e.g., by an institutional review board)?** No.

### A.5.4 Preprocessing/cleaning/labeling

**Was any preprocessing/cleaning/labeling of the data done (e.g., discretization or bucketing, tokenization, part-of-speech tagging, SIFT feature extraction, removal of instances, processing of missing values)?** In the original TERRA-REF project, the gantry system moved over the field and records its field position as it takes images. Our pre-processing of this original, raw TERRA-REF data was limited to cropping the original TERRA-REF images to rectangular regions that viewed one field plot (within which a unique cultivar was grown), and converting 3D line scanner voxel output to an image representation.

**Was the "raw" data saved in addition to the preprocessed/cleaned/labeled data (e.g., to support unanticipated future uses)?** The raw data is available through the original TERRA-REF collection [1].

**Is the software that was used to preprocess/clean/label the data available?** The raw TERRA-REF data is incredibly large and complex. The pre-processing code, similarly, is incredibly complex and tied in with the TERRA-REF processing pipeline, making it challenging to share only the pre-processing code that produced the data used in this benchmark. If anyone wishes to work with the raw TERRA-REF data, instead of our curated benchmark data, we are happy to provide pointers and make introductions to the TERRA-REF data processing team.

### A.5.5 Uses

**Has the dataset been used for any tasks already?** The SG×P Benchmark consists of binary classification tasks that relate to SNPs with both known and unknown genotype×phenotype relationships. The SNPs with known genotype×phenotype relationships were utilized in our prior work [3, 4].

**Is there a repository that links to any or all papers or systems that use the dataset?** Any papers or systems that use this dataset produced by the authors will be referenced on the `https://sorghumsnpbenchmark.com` website, and we encourage anyone using the dataset and benchmark to share their work on the Benchmark Forum at `https://forum.sorghumsnpbenchmark.com`.

**What (other) tasks could the dataset be used for?** While the SG×P Benchmark focuses specifically on binary classification of SNPs to understand the genotype×phenotype relationship in bioenergy sorghum, the data could be used for other plant phenotyping tasks, such as cultivar or subpopulation identification, leaf segmentation or flower detection. It also could be used for evaluation of more general machine learning models for multisensor fusion or time series understanding or prediction.

**Is there anything about the composition of the dataset or the way it was collected and preprocessed/cleaned/labeled that might impact future uses?** This dataset solely consists of imagery and genomic data for bioenergy sorghum grown in 2017 as part of the TERRA-REF project, which could impact generalization for other crops or environmental settings/conditions.

**Are there tasks for which the dataset should not be used?** No.

### A.5.6 Distribution

**Will the dataset be distributed to third parties outside of the entity (e.g., company, institution, organization) on behalf of which the dataset was created?** The dataset and benchmark are publicly available at `https://sorghumsnpbenchmark.com`.

**How will the dataset will be distributed (e.g., tarball on website, API, GitHub)?** The data is shared as CSV files for data splits and metadata and tarballs for imagery on the `https://sorghumsnpbenchmark.com` website.

**When will the dataset be distributed?** The dataset and benchmark have been available since June 6, 2023.

**Will the dataset be distributed under a copyright or other intellectual property (IP) license, and/or under applicable terms of use (ToU)?** Both the original TERRA-REF dataset and the derived datasets used in the SG×P Benchmark are released under a CC0 1.0 Universal (CC0 1.0) Public Domain Dedication license.

**Have any third parties imposed IP-based or other restrictions on the data associated with the instances?** No.

**Do any export controls or other regulatory restrictions apply to the dataset or to individual instances?** No.

### A.5.7 Maintenance

**Who will be supporting/hosting/maintaining the dataset?** Support, hosting and maintenance for the dataset and benchmark will be provided by the project team at Saint Louis University.

**Is there an erratum?** Should any errors be discovered in the dataset and benchmark, they will be publicly disseminated on the `https://sorghumsnpbenchmark.com` website.

**Will the dataset be updated (e.g., to correct labeling errors, add new instances, delete instances)?** Barring the discovery of any errors, the dataset and benchmark will not be updated .

**If the dataset relates to people, are there applicable limits on the retention of the data associated with the instances (e.g., were the individuals in question told that their data would be retained for a fixed period of time and then deleted)?** N/A

**Will older versions of the dataset continue to be supported/hosted/maintained?** N/A

**If others want to extend/augment/build on/contribute to the dataset, is there a mechanism for them to do so?** The CC0 1.0 Public Domain Dedication license allows anyone to utilize the data shared as part of the SG×P Benchmark for any purpose.

### A.6 Author Responsibility Statement

The authors of this work bear all responsibility in case of violation of rights, etc., that result from the use of this dataset and benchmark.

### A.7 Hosting, Licensing and Maintenance Plan

The SG×P Benchmark website (leaderboards and forums) are hosted via DigitalOcean. The underlying data is hosted on servers at Saint Louis University, and will additionally be archived on Data Dryad (`https://datadryad.org/stash`). The data is released under a CC0 1.0 Public Domain Dedication license. The authors commit to the maintenance and public availability of these platforms and data for at least five years from publication data.

## Appendix References

[1] Maxwell Burnette, Rob Kooper, J. D. Maloney, Gareth S. Rohde, Jeffrey A. Terstriep, Craig Willis, Noah Fahlgren, Todd Mockler, Maria Newcomb, Vasit Sagan, Pedro Andrade-Sanchez, Nadia Shakoor, Paheding Sidike, Rick Ward, and David LeBauer. "TERRA-REF Data Processing Infrastructure". In: *Proceedings of the Practice and Experience on Advanced Research Computing*. Ed. by Sergiu Sanielevici. ACM, 2018, 27:1–27:7. DOI: 10.1145/3219104.3219152.

[2] Timnit Gebru, Jamie Morgenstern, Briana Vecchione, Jennifer Wortman Vaughan, Hanna Wallach, Hal Daumé Iii, and Kate Crawford. "Datasheets for datasets". In: *Communications of the ACM* 64.12 (2021), pp. 86–92.

[3] Abby Stylianou, Robert Pless, Nadia Shakoor, and Todd Mockler. "Classification and Visualization of Genotype × Phenotype Interactions in Biomass Sorghum". In: *Proceedings of the IEEE/CVF International Conference on Computer Vision (ICCV) Workshops, Computer Vision Problems in Plant Phenotyping and Agriculture Workshop*. Oct. 2021.

[4] Zeyu Zhang, Madison Pope, Nadia Shakoor, Robert Pless, Todd C. Mockler, and Abby Stylianou. "Comparing Deep Learning Approaches for Understanding Genotype x Phenotype Interactions in Biomass Sorghum". In: *Frontiers in Artificial Intelligence* 5 (June 2022).