# OpenReview forum: "SG×P : A Sorghum Genotype × Phenotype Prediction Dataset and Benchmark"
_NeurIPS.cc/2023/Track/Datasets_and_Benchmarks — NeurIPS 2023 Datasets and Benchmarks Poster_

### Official Review · Reviewer_w226 · 2023-07-21
**Good paper, requesting minor revision before acceptance.**

**Rating:** 6
**Confidence:** 4
**Clarity:** The paper is well written.

**Strengths:**

The dataset is incredibly unique and contain rich, fine-grained information for target plants, with RGB, 3D, and SNP data. The availability of data is good, with a dedicated webpage for the dataset. The dataset is available for commercial and academic usage under the Creative Commons Attribution 1.0 license, making it ideal for a wide range of research and applications in academia and industry.

**Additional Feedback:**

Please work more on the formatting. The manuscript does not seem to use the correct format for submission. Especially, the line numbers are missing. Some references are provided in parenthesis and bracket; e.g., ([1]). Please provide proper formatting for the manuscript.

Because NeurIPS targets a wide range of audiences, a more detailed explanation and reasoning behind the selection of SNPs and reference genome appears necessary; for example, what is SNP and why was it selected for this dataset? Why wasn't the whole-genome sequencing data shown instead? Why does this paper focus on homozygous cases rather than heterozygous cases? Why was BTx623 chosen as the reference genome? I understand the page limit, but I think the authors can move figure 4 and section 6 texts to supplementary and leave necessary components in the main text. Instead, please provide detailed overview of SNP and the reasoning behind selecting the marker in section 6. The paper would appeal to broad range of audiences if genomic-level data selection is better explained.

In page 3, reference 26 appears to be link to a Kaggle competition. It would be better to cite a publication about plant classification (https://doi.org/10.3389/fpls.2021.787127) for the previous year's competition that addresses similar problem.

The current link for the pre-train model does not work.

While the code is readily available, a better detailed github repository would greatly aid in making this effort reproducible.

**Correctness:**

Generally yes. Some claims are not correct; for example, pre-trained model is not available.

**Documentation:**

The dataset could benefit from a more thorough walkthrough of the data and code instructions from the github repo.

**Ethics:**

There are no ethical concerns.

**Limitations:**

The main limitation of this dataset is constraint of genomic-level information to SNP. Better reasoning of the use of SNP compared to other genetic markers would increase the quality of this work. Release of the whole gene sequence data would add more value to this dataset, but not required.

**Opportunities For Improvement:**

First of all, the manuscript does not seem to use the correct format for submission. Especially, the line and page numbers are missing. Please provide proper formatting for the manuscript.

Because NeurIPS targets a wide range of audiences, a more detailed explanation and reasoning behind the selection of SNPs and reference genome appears necessary; for example, what is SNP and why was it selected for this dataset? Why wasn't the whole-genome sequencing data shown instead? Why does this paper focus on homozygous cases rather than heterozygous cases? Why was BTx623 chosen as the reference genome?

The current link for the pre-train model does not work.

While the code is readily available, a better detailed github repository would greatly aid in making this effort reproducible.

**Relation To Prior Work:**

Sufficient.

**Summary And Contributions:**

This study provides a first-of-its-kind dataset and benchmark for linking visual plant phenotypic characteristics to genetic information. The contribution of this paper is apparent, as it is the first-ever plant phenotype-genotype dataset that includes all RGB, 3D, and genetic information. It contains 543,574 RGB and 536,153 3D images that overlap with 15 and 2,717 SNPs with known and unknown phenotypic expressions, respectively. The dataset is released under Creative Commons Attribution 1.0, which makes it free for commercial and research use, possibly enabling for a wide range of applications in domains such as computer vision, molecular biology, plant science, and agriculture. One major limitation of this study is the availability of genomic-level data. The study claims that full genome sequencing was performed; however, entire DNA sequence information is not included in this dataset; instead, only SNPs are included for the genomic-level data, which may limit studying fine-grained phenotype-genotype relationships that could lead to novel discoveries. The release of the full genome sequencing data may open up new research avenues for encoding DNA sequence data in the context of machine-recognizable phenotypic characteristics. However, I don't feel that it is necessary for this particular study, since the paper addressed their claim sufficiently with the SNP data. Nonetheless,  more details on the SNPs is necessary to appeal for general audience in AI. Considering the venue of NeurIPS —  whose audience is not limited to molecular biologists — it would make the manuscript much suitable for this conference (and increase the overall impact) with more detail regarding the nature of the presented type of genomic data (i.e., single nucleotide polymorphism) and the rationale behind the selections. In addition, the manuscript contains minor formatting issues which needs to be corrected. Finally, the dataset can be benefitted by better-explained github repository, as the baseline reproducibility requires a sufficient walkthrough of the data and instructions for executing the codes. After addressing these issues, I would be happy to accept this paper with higher ratings.

---

> ### Author Response · Authors · 2023-08-21
>
> We thank reviewer w226 for their time and consideration in constructing their review! We have rephrased the reviewers questions/concerns and listed our responses below:
>
> _Q: No line numbers?_
> We apologize that we did not include line numbers in the template. Since the benchmark track is not generally anonymized we used the final copy NeurIPS template and did not catch that this disabled things like line numbers. We re-enabled this in our updated version.
>
> _Q: Why SNPs and not full genome? Why homozygous? Why BTx623? Why these markers?_
> We define what SNPs are and their relation to the overall genotype on lines 162-166. For related genomes, SNPS are the places where those genomes differ – so the SNPs are the relevant differences between those genomes. We only consider homozygous cases in part because it makes the classification task simpler and because the phenotypes expressed by heterozygous plants are often far less consistent and more varied from one plant to the next. We use BTx623 as the reference genome because that is the standard in sorghum literature. The reason the sorghum research community uses that is because it was the first well-sequenced sorghum line. We explain the choice of markers used in Section 6 on lines 189-199.
>
> _Q: Fix Herbarium reference to paper instead of Kaggle_
> We have corrected this – thank you for pointing it out!
>
> _Q: Broken links for pretrained model_
> The broken links have been fixed on both the sorghumsnpbenchmark website and the github repo!
>
> _Q: Improved documentation on github_
> We have updated the github repository to have significantly more thorough documentation and instructions.  We have also added helper functions (and documentation) for working with the dataset files and labels.

---

> > ### Author Response · Authors · 2023-08-28
> > **Request for ongoing discussion**
> >
> > Hi Reviewer w226!
> >
> > Thank you again for your thoughtful review.  We very much appreciate your comments and have edited the paper and the competition website in response; I think both are better now.  If you have any further questions or comments, please let us know!

---

> > > ### Comment · Reviewer_w226 · 2023-08-31
> > >
> > > Thank you for the response. Score was increased.
> > >
> > > >*Fix Herbarium reference to paper instead of Kaggle* We have corrected this – thank you for pointing it out!
> > >
> > > I think the main text talks about herbarium 2022, which has no published work associated. Please note that the suggested reference paper is about kaggle competition from previous year (herbarium 2021).

---

### Official Review · Reviewer_XBbG · 2023-07-21
**A large, interesting, and well-curated dataset but the associated materials on the website need more attention before they are ready for release**

**Rating:** 7
**Confidence:** 3

**Strengths:**

This is a rich and interesting dataset with the potential to drive the use of machine learning to make interesting discoveries about previously undiscovered phenotypes expressed by SNPs. I am not familiar with with the TERRA-REF project from which the dataset is drawn, so I’m unable to make a judgement as to how this dataset differs from other related datasets released from the same project, but the dataset and problem as stated are interesting an appear to be very relevant to the community.

**Additional Feedback:**

Just a thought - no need for action if you don't think it's useful:

The per-SNP annotations are provided in a file format which makes the dataset much larger than it needs to be, which might hinder accessibility of the dataset as well as driving up the cost of serving the dataset if it becomes popular. The total size of the annotation file is around 52 Gb compressed, and it uncompresses to over 200Gb. If I understand correctly, the per-SNP annotations give a binary class for each image, but not all images necessarily have a binary label for each SNP. Given that there around 1M images across both modalities, and for each SNP, each of these will either have an annotation of either 1, 0 or not in dataset, the complete dataset could be represented as a mapping from a 32-bit integer ID for an image to a bitmask with 2 bits per SNP to represent the possible annotations. For 3000 SNPs, this would lead to a data size of 6032 bits (754 bytes) per image. For 1M images, this would be around 750Mb before any further compression, which is over 70 times smaller than the distributed size of the data. The authors should consider formatting the data in such a way that the download is smaller and providing scripts to generate the annotation files in their current form as needed.

**Clarity:**

Overall the paper is very clear and well-written.

In section 4, I recommend the authors clarify what metric is reported for model performance in table 3. I assume it’s classification accuracy on the test set, but this is not explicitly stated anywhere.

In section 6, the authors present details of their website including the leaderboards and community forums. An example of a forum interaction is provided, with what I viewed as an implication that there is already a community of researchers actively contributing to the website. On closer inspection, it transpires that this is in fact the only post on the forum, and the interactions appear to be between two of the authors of the paper. While this is clearly a new site, and the authors (and I) naturally hope that the community will thrive in time, I find this presentation to be slightly disingenuous, and I would suggest that the authors consider rewording this section to make the current state of the website clear.

**Correctness:**

The dataset construction appears sound, and is well described in the paper. A full description of the baseline model is provided, along with code to recreate it.

**Documentation:**

Hosting, licensing and maintenance plans are referenced in the supplementary material and seem reasonable.

There is no documentation provided in the GitHub repository linked from sorghumsnpbenchmark.com. This should be addressed, as the lack of a README for the code prevents someone from easily rebuilding the model.

There is no licence provided for the code on GitHub. This should be addressed.

The link to the pretrained baseline model provided https://sorghumsnpbenchmark.com/ is broken.

The link to this paper from https://sorghumsnpbenchmark.com/ is currently broken. I believe Openreview allows for public linking of papers under review, so the paper could be linked to already if the authors are comfortable to do that.

How the data is structured within the download tarball is not specified anywhere on the website. The authors should consider adding documentation of the data structure and also making a small example dataset available for development and documentation purposes. The need to download a full dataset in order to understand the structure of the available data may be a barrier to researchers getting started on the problem.

The main data tarball does not contain the images. As far as I can see, the images referenced in the paper are not in fact currently available from the website. This needs to be rectified, as they are a critical part of the dataset.

There is a separate file that contains the data used to pre-train the baseline model. It is not immediately obvious from the website whether this file contains the original images referenced in the paper or not, but section 4 of the paper states that these pre-training images are in fact from a different, 2019, dataset. It would be useful to provide more information on what’s in each downloadable file on the site.

The dataset is currently only available as unversioned tarballs. The authors should consider using a data hosting platform like Zenodo which allows for supported hosting of versioned datasets with associated DOIs for each version.

**Ethics:**

No obvious ethical concerns. The dataset contains RGB and false-colour images of crops, with associated binary labels for specific genotypes (SNPs). There is little obvious scope for social or environmental harm from this dataset.

**Limitations:**

One limitation of the dataset is that the 3D and RGB images are not completely temporally and spatially aligned. The authors specifically address this limitation in section 3.2 and explain their rationale for choosing the closest possible image pairs. This rationale seems appropriate and reasonable, and the approach is made very clear.

I don’t see any other clear limitations which the authors have not addressed.

**Opportunities For Improvement:**

The RGB and 3D images referred to in the paper do not appear to be available from the website linked in the paper. Annotations are available, and training data for the baseline model is available (based on a different set of images from 2019), but I can’t find the actual images for download anywhere. This needs to be rectified. If the images are in fact available for download from the website, then this needs to be made very obvious on the page, as I couldn’t find them. If they’re not, they need to be added.

Practical concerns:
The leaderboard appears to allow for users to directly upload their own scores for a particular task https://sorghumsnpbenchmark.com/submit. Do the authors have a plan for how they intend to validate and weed out spurious results and spam in the leaderboards?

**Relation To Prior Work:**

The Introduction and Background sections cite prior work and place the dataset in context. There is however perhaps scope here to be more explicit about this dataset relates to others already available. I suspect that this dataset is the first of its type (real world images of a range of different crop varieties with associated genotyping information) if so the authors should call this out explicitly, and if not then other similar datasets should be highlighted.

**Summary And Contributions:**

The authors present a dataset of over 1M images (around half being colour (RGB) and half false colour 3D line scan images) of sorghum plants in a field setting, along with associated binary annotations for subsets of the images for each of 15 single nucleotide polymorphisms (SNPs) with known phenotypic expression and a further 2717 SNPs with unknown phenotypic expression.

The paper presents a website containing self-reported leaderboard scores for model performance on each individual SNP, along with links to the data for download, either all together or separately per SNP. The website also hosts a forum for discussion of results.

The authors provide a baseline model which is used to generate features for subsequent SNP classification tasks. Features generated from the baseline model are used to train classifiers. These results for the known SNPs are presented in the paper, and baseline results for all SNPs are available on the website.

---

> ### Author Response · Authors · 2023-08-21
>
> We thank reviewer XBbG for their thoughtful review. In the interest of space, we have rephrased the reviewers questions/concerns and listed our responses below:
>
> _Q: Specific metric not stated?_
> We have added this clarification in Table 3 and in the text.
>
> _Q: Example discussion was not clearly between authors?_
> We appreciate this feedback. We had attempted to make it clear that this was an example of the type of discussion we envision, but have revised the text to make this more apparent. Our plan is to broadly disseminate the benchmark following its publication – both to the ML community through NeurIPS itself, through relevant ML communities (such as the AI for Science Workshops at NeurIPS, the Agriculture-Vision and EarthVision Workshops at CVPR, the Computer Vision Problems in Plant Phenotyping and Agriculture Workshop at ICCV and ECCV, the ICML Workshop on Computational Biology, and others), and also to data-science and ML-inclined biology researchers, such as those that are active on the SorghumBase user group.
>
> _Q: Broken links?_
> These have been fixed! We thank you for pointing them out!
>
> _Q: Data structure not detailed?_
> We have updated the website to include additional documentation on what is included in each file.  We have also added an example dataset that contains images, metadata, labels and code for a subset of the data (1000 RGB images, 1000 3D images, and 3 markers) so that participants can explore the benchmark without having to download the entire dataset, and updated our GitHub repo to include helper functions for downloading and working with specific SNP datasets.
>
> _Q: No images?_
> The lack of link to the images was an accidental oversight – we have added the link, as well as additional documentation on what is included in each file. We have also added an example dataset that includes images, metadata, labels and code for a subset of the data (1000 RGB images, 1000 3D images, and 3 markers) so that participants can explore the benchmark without having to download the entire dataset.
>
> _Q: Post on Zenodo for versioned datasets?_
> We appreciate the recommendation to host on a platform like Zenodo. We opted to host on an academic website due to size and budget constraints, but Zenodo is an excellent option. Our dataset (and even most of the individual SNP-specific datasets) are larger than Zenodo’s default file limit, but we have reached out to them about the possibility of hosting the full dataset there (they sometimes make exceptions to the file size limit).
>
> _Q: The label files could be significantly smaller?_
> We appreciate this suggestion! We’ve created a new version of the per-SNP annotations following the reviewer’s approach, and it is 633MB. We’ve put this new annotation file on the sorghumsnpbenchmark website and have provided code to work with this file.

---

> > ### Author Response · Authors · 2023-08-28
> > **Request for ongoing discussion**
> >
> > Hi Reviewer XBbG,
> > Thank you again for your thoughtful review; we've used your comments to fix some problems on the website, and tried to clarify some of our design choices in response to your comments.  If you have any other questions or additional feedback, please let us know!

---

> > ### Comment · Reviewer_XBbG · 2023-08-28
> > **Significant improvements to the dataset**
> >
> > Thank you for all the updates.
> >
> > I'm pleased that the suggestion on reducing the size of the annotation dataset worked so effectively. The added documentation and work on automatic data loaders that has been done in the GitHub repository is very useful and makes the dataset much more accessible.
> >
> > I'm happy to increase my review score in light of these updates.

---

### Official Review · Reviewer_J6ZQ · 2023-07-22
**Well-constructed and extremely interesting dataset**

**Rating:** 8
**Confidence:** 4
**Clarity:** Yes, it is very well written and easy…

**Strengths:**

There are many positives to this submission:

* The dataset provided here is extremely rich in imagery and genetics, providing many opportunities for vision and multimodal research.
* The data is extremely permissively licensed (CC0).
* The leaderboard and forum enable engagement from researchers working on the dataset.
* Improving crop yields and resistance to climate-induced stressors is a problem of great importance.
* The tasks are clearly described, and data is described in a way that is accessible to non-geneticists.

**Additional Feedback:**

N/A

**Correctness:**

The dataset is carefully constructed and fully available. The associated benchmark is reasonable though on the website could be better described.

**Documentation:**

In general the website and documentation are good. See improvements section above for how to improve the documentation.

**Ethics:**

No concerns.

**Limitations:**

See above. The other limitations raised are good to highlight and reasonable, and certainly do not preclude publication.

**Opportunities For Improvement:**

* Accessing the data is cumbersome. At this point everything is stored on https://cs.slu.edu/~astylianou/neurips_sorghum_dataset/ including the 230 GB .tar.gz file of all images. Hosting on one or more Cloud providers would simplify access for many.
* The authors indicate that all benchmark tasks are single-SNP only. Is the data organized in a way that the genome of individual plants can be pieced back together? This would enabled multimodal imaging+genomic prediction. If not, why not?
* Some terminology and data sizes are not clear. The authors mention ~300 cultivars and ~543k images. Does this mean that there are ~300 genomes? A bar plot of images per cultivar would also be helpful to understand the distribution of data.
* There is no link in the website to the images.tar.gz file, I had to hunt for it. A table from download link to a short description of its contents would be very helpful. Also, after download a description of the file contents and pointers to tools to analyze the genetic data would also be helpful.
* To upload to the leaderboard, you input everything including the evaluation metrics (top image accuracy, top cultivar accuracy). To ensure the leaderboard actually reflects reality, it would be better for individuals to upload their marker, image type, and then the set of predictions, and have the server compute the associated top image and cultivar accuracy values.

**Relation To Prior Work:**

Yes, section 2.2 provides comprehensive comparison to other plant phenotyping work.

**Summary And Contributions:**

The authors introduce http://sorghumsnpbenchmark.com, which includes a dataset of paired genome data and time sequence imagery with multiple modalities taken across an entire growing season, for over 300 sorghum varieties. All data is fully available and the website includes a leaderboard for evaluation and discussion forum for users. A set of strong baselines are also included.

---

> ### Author Response · Authors · 2023-08-21
>
> We very much appreciate the positive review! We are excited to clarify some of your questions below.
>
> _Q: Accessing data is cumbersome?_
> We are looking into hosting the full dataset and example datasets on Zenodo (the full dataset is larger than their default upload limit, but they are willing to discuss hosting larger datasets with researchers). We have also added an example dataset so that researchers can explore the data without having to first download everything, and have added code for downloading both the example dataset and the overall dataset to our GitHub repo.
>
> _Q: Do cultivars mean genomes? Dataset distributions unclear?_
> Yes – cultivars refer to different lines of sorghum with distinct genetics. SNPs are locations where these genetics differ. To help clarify dataset distributions, we have replaced Table 2 with a new Figure better showing the distributions per SNP.
>
> _Q: No images?_
> We accidentally only linked to the metadata and not the images. We have fixed this on the website and added additional documentation about what different files include on the benchmark website.
>
> _Q: People can submit their own scores?_
> Please see the response we posted to all reviewers addressing why we made this choice.
>
> _Q: No images?_
> The lack of link to the images was an accidental oversight – we have added the link, as well as additional documentation on what is included in each file. We have also added an example dataset that includes images, metadata, labels and code for a subset of the data (1000 RGB images, 1000 3D images, and 3 markers) so that participants can explore the benchmark without having to download the entire dataset.
>
> _Q: Reviewers can submit their own scores?_
> Please see the response we posted to all reviewers addressing why we made this choice.

---

> > ### Author Response · Authors · 2023-08-28
> > **Offer for ongoing discussion**
> >
> > Hi, Reviewer  J6ZQ!
> >
> > Thank you again for your comments; we've made some changes to clarify the text in response to your comments and explained some of our design choices in answer to your questions.  We appreciate the opportunity to have this conversation so if there is anything else we can answer, please let us know!

---

### Official Review · Reviewer_aqR6 · 2023-07-22
**Review of SGxP dataset**

**Rating:** 5
**Confidence:** 4
**Clarity:** The paper is overall well written and…

**Strengths:**

- The authors did a good job explaining the plant science background necessary to understand the dataset in a concise way that doesn't overwhelm the reader with domain-specific details.
- The paper is overall well organized and easy to read.
- I like the idea of providing a discussion forum to try to promote collaborative discussions between users and beneficiaries of the benchmark, and I appreciated the anecdote about a researcher from their team using the discussion forum in the way it was intended.

**Additional Feedback:**

Would it be helpful to provide the 3D images in their original voxel format to enable exploration of model architectures that can operate on 3D images?

**Correctness:**

To my knowledge, the claims are correct. The dataset is constructed in a sound way (I appreciated that the authors attempted to avoid data leakage by keeping cultivars/plot images within the train or test set, not split between the two). See earlier comments for discussions of the limitations of the dataset.

**Documentation:**

There seems to be sufficient documentation, though information about spatial sampling should be provided.

**Ethics:**

I do not see any ethical concerns with the dataset.

**Limitations:**

- The authors do point out the limitations of the dataset being constrained to a single crop in a single season in a single field site. While this constrained dataset could still be useful, it's not clear how the models or dataset could be useful beyond this single dataset. The impact could be more convincing if it were clear that there was a large community of researchers who would be ready to interact with this discussion forum beyond their research group (or the discussion forum connected to some existing community of researchers).

**Opportunities For Improvement:**

- There seems to be a large leap of faith required to go from "binary classification of reference vs. alternate SNP" to "improved understanding of genotype-phenotype relationships". It's not clear how the classification task would lead to transformative understanding of these relationships. It seems that this link rests entirely on observations of the relevant phenotype characteristics being highlighted in class activation maps, but 1) CAMs should be interpreted with caution (see [1] below, for example), and 2) without seeing examples of this applied to unknown SNPs where the phenotype expression you are looking for is unknown, this could simply be confirmation bias (i.e., the phenotype is known to be waxy, the CAM confirms it).
- While there are many images in the dataset, the diversity of those images is not clear. A picture or map of the field site that shows how images are sampled from the area to show how much overlap or diversity there is between images would be helpful. For example, how many images are taken from the same plot which are probably pretty similar to each other? The TERRA-REF website says the plants cover 1 acre, so this suggests there might be a lot of similarity/autocorrelation between images in the dataset.
- I did not see details about how many images were used for pre-training the Resnet-50. Given previous work that shows ImageNet-pretrained models are useful initial weights for very different finetuning tasks, I was surprised to see only results from a custom pretrained model and not an ImageNet pretrained model which may have done better. In general, one architecture baseline seems limited.
- In addition, the baseline results were only provided for the 15 known SNPs but none of the unknown ones. Why? Aren't these the datasets the authors want to see researchers use most? (They seem to be on the website, but were not discussed in the paper.)

[1] Rudin, C. (2019). Stop explaining black box machine learning models for high stakes decisions and use interpretable models instead. Nature machine intelligence, 1(5), 206-215. https://www.nature.com/articles/s42256-019-0048-x

**Relation To Prior Work:**

The authors gave a good description of existing datasets in plant science and phenotyping, and discussed how their dataset differs and is similar to existing datasets.

**Summary And Contributions:**

The authors present a new benchmark dataset for predicting whether a plant in an image (RGB, 3D, or both) has the reference or alternate version of an SNP (single nucleotide polymorphism). The dataset includes 15 SNPs with known genotype-phenotype relationships and 2,717 SNPs with unknown relationships. The authors hypothesize that models that learn to accurately predict reference vs. alternate classes will learn latent features useful for understanding genotype-phenotype relationships. The authors present results from a baseline ResNet-50 model pre-trained on field plots and fine-tuned for each SNP dataset with RGB only, 3D only, and RGB+3D inputs. They also provide a website hosting the dataset and leaderboard, with a discussion forum to encourage interdisciplinary collaboration between machine learning and agriculture researchers.

---

> ### Author Response · Authors · 2023-08-21
>
> We thank reviewer aqR622 for their thoughtful review. In the interest of space, we have rephrased the reviewers questions/concerns and listed our responses below:
>
> _Q: Why CAMs?_
> We present CAMs as one possible visualization to help discover “what the models are focusing on” – but do not mandate any particular explainability approach. The ML experts and biology experts on this team have been exploring the relationship between SNPs and their expressed phenotypes, and have explored a variety of different explainability approaches, including not just CAMs and gradient based heatmaps, but also montages and generative modeling approaches. Different approaches are relevant for different tasks and we are excited to see what participants share during discussions with biologists on the SGxP discussion boards!
>
> _Q: Image diversity?_
> We are confident that there is sufficient variability in the images to support this task. One reason that we train our pretrained model on plot ID instead of cultivar ID is so that we can verify the model is able to learn generalizable and biologically meaningful features – we observed that images from both of the two plots from the same cultivar have similar features extracted from the learned model (even though there was no loss that explicitly ensured this), indicating that this pretrained model has learned generalizable and biologically discriminative features from the images provided.
>
> _Q: Pre-training dataset size and comparison with other pre-trained models?_
> We have added details on the size of the pre-training dataset to Section 4, and have found in prior experiments on SNP classification that our pretrained model outperforms an ImageNet pretrained model [69]. We include only one architecture due to the compute restraints when training over 8100 different models for the baseline (2717 markers x 3 modalities). We believe this is sufficient, as the goal of the baseline is simply to provide participants with a reasonable point of comparison (“how well would a fairly simple task-specific model perform and am I doing better than that?”), rather than exhaustively attempting to find the best possible model.
>
> _Q: Why only baseline results for known SNPs?_
> Section A.3 shows aggregate performance of the baseline on the unknown markers (mean, std, 25th, 50th and 75th percentile performance per modality, by-image and by-cultivar). These aggregate statistics are less important for participants, who will focus on performance for specific markers and modalities – it is infeasible to include each of those in the paper, and instead is better presented in detail on benchmark web interface.
>
> _Q: What researchers will use this?_
> There is a large and vibrant community of sorghum-specific researchers! One particular community that the authors are active in and will work with when seeking participation is SorghumBase (https://www.sorghumbase.org/) which has an active community of thousands of sorghum researchers, with active discussion boards and mailing lists. There are also large and active populations of machine learning researchers that are excited to engage in ML for science -- see our top level comment regarding this.
>
> _Q: Why processed 3D images?_
> We provide processed 3D images in large part due to size constraints – the raw 3D data in voxel format would be absolutely massive. A single raw 3D data file is over 8MB – if we shared these for every 3D data point in the benchmark, it would be over 4TB. This data is, however, available as part of the original, publicly available TERRA-REF website and we allow participants to utilize external data as they see fit in the benchmark (we have updated the paper to make this more clear).

---

> > ### Author Response · Authors · 2023-08-28
> > **Request for discussion**
> >
> > Hi, Reviewer aqR6!
> >
> > Thank you again for your comments.  We are very excited to share this dataset, and tried to explain some of our design choices in answer to your questions.  We appreciate the opportunity to have this conversation, so if there is anything else we can answer, please let us know!

---

### Official Review · Reviewer_VdiJ · 2023-07-25
**A worthy goal, but I'm unconvinced these are the 8196 tasks to achieve it**

**Rating:** 6
**Confidence:** 4
**Correctness:** 1. For each SNP, the dataset has part…

**Strengths:**

1. In regards to the ethical and social implications, this is a strong positive point for the dataset. A better understanding of the relationship between sorghum genotypes and phenotypes would clearly be of societal benefit.

2. The dataset has clear relevance to those interested in studying genotype/phenotype relationships, especially in the context of plants.

3. The dataset combines RGB images, 3D scans, and DNA information --- an interesting combination of modalities. There are over half a million RGB images and half a million 3D scan images, which is a substantial amount of annotated data.

4. Using binary tests with balanced test sets makes it simple to implement an evaluation and understand the resulting accuracy of an individual test.

**Additional Feedback:**

In my opinion, Figs 3, 4 and §6 don't add significant value to the paper and I would recommend removing them entirely and spending the space this frees up on something more useful.

It would have been helpful if the authors had enabled line numbers in LaTeX, as then I would have been able to refer to typos etc by line number.

In summary, I think the purpose of the dataset is good, but I am not convinced that the way the benchmark is structured will allow it to uncover the answers the authors are looking for.

**Clarity:**

The paper is generally well written, using clear language without grammatical errors.

Although some features of sorghum are mentioned, and the uses of some of the species within its family are referenced later in the paper, a description of sorghum is not directly given in the introduction. It would help the machine learning audience understand the context if this was included. Something like this (my own suggestion based on Wikipedia and the details present in the paper, just so you have a sense of what sort of context I would have liked to have gained from reading the introduction):
> Sorghum is a genus of flowering plants in the grass family. The 25 species in the sorghum genus have a variety of anthropogenic uses, including the harvesting of cereals and grains for human consumption, fodder plants for livestock, bristles for brooms, and biofuels. Sorghum crops are grown across the world, but form an especially important staple to the peoples of many arid and semi-arid regions due to its drought- and heat-resistance properties.

### Typographic errors
- p2, §1: "understanding the ~the~ SNPs" repeated word
- p2, §2.1: "crucial area of research in plant biology ~(~[6, 7, 13, 39, 47]~)~" doubled-brackets around citations are superfluous (repeated around many citations in the paper)
- p3, §2.2: "information,[54]" missing space
- p3, §2.2: "sorghum genome , or," extra space
- p4, §3.1: "Figure 2(right)," missing space (also p5/§3.2, p9/§6)
- p4, §3.1: "buildup" -> build-up
- p7, §4: "balance sampler" -> balance**d** sampler
- p7, §5: "straight forward" -> straight-forward
- p7, §5: "~to~ for the SNPs" -> for the SNPs
- Table 2: Cells should be right-aligned so the stdev and mean numbers have their digits aligned,

### Citation errors

Some incorrect casing in citations:
- [58] "inaturalist"
- [61] "biosystems engineering"
- Some citations are in title-case, others are in lower case.

Some errors pertaining to DOIs:
- [14] Use DOI field instead of a URL pointing to a doi.org URL; eprint is redundant with DOI
- [30] URL is redundant with DOI
- [40] DOI field contains a URL; eprint is redundant with DOI; URL is redundant with DOI
- [63] DOI field contains a URL; URL is redundant with DOI
- [67] DOI field contains a URL; eprint and URL are redundant with DOI

**Documentation:**

1. I successfully downloaded `genetic_marker_datasets.tar.gz` from the website with the URL given in
> To download the entire SGxP dataset(for all SNPs), click here

and found it contained 16,394 CSV files describing the train/test partitions for 2,732 SNPs. Although the CSV files contain the names of JPG images, I was unable to work out how to download the images. Perhaps this is because the images need to be downloaded separately from the TERRA-REF project? I checked on https://terraref.org/ even though it was not linked to on https://sorghumsnpbenchmark.com/ or in the paper, and it was not clear how to download the images from there either. I hope my inability to download the images is because I missed something on the website, but this seems unlikely. I have seen that there are RGB and 3D scan images in the file used to pretrain the baseline model, `genetic_marker_pretrain_dataset.tar.gz`, but it is not clear that these images are strictly the same as or a superset of the images that comprise the actual dataset. Certainly if this is where the imagery is to be found, this is not an appropriately indicated download link. Obviously, being able to download the images is essential to the validity of the dataset and if the images comprising the dataset are not accessible, this paper is a trivial reject. To make my review score meaningful, I have scored it assuming the images are in fact downloadable and the current documentation is just poor at explaining how to download the images. I expect the authors to make an update to the documentation on their website for the dataset so it is clear how to go about downloading the images. If I am still unable to understand the instructions and download the images myself at the end of the author-reviewer dialogue period I will downgrade my review score accordingly.

2. The code on the github page does not have any documentation, making it impractical to replicate the experimental baseline. I briefly looked at the code anyway and found that train.py has an `if __name__ == '__main__':` block which is block commented out by converting it to a multi-line string. This indicates to me that the training code has not been shared in an executable state.

**Ethics:**

No.

**Limitations:**

1. Some SNPs might have no phenotypic impact but be correlated with another SNP that does have a phenotypic output. For example, a mutation at SNP #123 might co-occur with a mutation at SNP #2144 because they both happen to occur in the same cultivars. These correlations will cause spurious conclusions about of the impact of some of the SNPs. (This limitation is not acknowledged in the paper.)

2. Once a model achieves a high accuracy, the task of determining which phenotypic features it has detected is non-trivial. Although having a model achieve high-performance at classifying an SNP is a step towards determining the phenotypic changes that the SNP presents, determining what the phenotype is from this model is a significant task. The complexity of this is not really acknowledged in the paper --- trying to go from a performant model to the phenotype that is presented is a serious undertaking that will require the manual labour of a data scientist; it will have to be applied to selected models on selected SNPs only a handful of times, and can not be automatically rolled across the entire dataset of thousands of SNPs. In order to maximize the utility of this manual post-hoc model analysis, it is important that the benchmark is very good at finding models and SNPs that are most worthy of manual investigation.

3. Some genotypes may result in phenotypic changes which are not visible and hence can not be discovered by this dataset. This may include phenotypes of high interest, such as changes to the root structure that improve the aridity-resistance of the plant. (This limitation is not acknowledged in the paper.)

4. While the goals are noble, I do not anticipate ML researchers participating in the discussion forum. The discussion forum may add value to biologists trying to understand sorghum SNPs, but not as much to ML researchers, hence I do not expect them to participate.

5. In general, whilst there is value to the sorghum biology community, I'm struggling to see the utility of this dataset to machine learning researchers. Why would an ML researcher use this dataset instead of an existing fine-grained image classification dataset? Just because ML can be applied to a dataset does not inherently mean that dataset is of interest to the ML community (and the ML community is the target audience of this conference track).

**Opportunities For Improvement:**

1. In general, the same SNP can have multiple viable mutated (alternative) forms (e.g. the letter A could be changed to G, C, or T; or it could be duplicated, or deleted; yielding up to 5 possible changes at that SNP). However the paper describes the task as a series of binary classification tasks. It is not clear if *all* pertinent SNPs in sorghum only have one viable alternative form, or if some SNPs have multiple mutants and these are each considered their own binary task (e.g. one task is SNP #123 being changed A->G and another task is SNP #123 being changed A->C). If some of the 2732 SNPs have multiple mutants, would it not be better to have a multi-class classification task instead? Some clarification on this would be helpful.


2. Table 2: Knowing the mean and stdev is not very helpful to the reader. It would be more useful to have histograms showing the distribution of the number of samples for the SNPs. The distribution is likely to follow a power-law, in which case the median number of samples across tasks is nowhere near the mean, which is inflated by the more populous classes. *Update:* I checked myself and the median is 27131 samples per class, with a minimum of 9490.


3. In §3.2, it would be very helpful to see the distribution of class (im)balance across the SNPs, or a quantification of the class imbalance (as is seen in the [iNaturalist paper](https://arxiv.org/abs/1707.06642), for instance), rather than an unqualified statement that there is significant class imbalance in the training sets.


4. p6, §4:
> we then take this pre-trained model and train a single fully connected layer with two outputs, corresponding to the Reference and Alternate labels

It is unclear why two output logits are used instead of one. Using two output logits is overparameterized since a binary classification task needs only one logit.


5. In §4: Since the pretraining task is not directly aligned with the downstream classification tasks, it would be great to see the performance with some fine-tuning as well, not just when using a frozen encoder.


6. Table 3: Caption says "Performance" without specifying the metric used. Presumably this is accuracy, since it is a binary classification task with balanced test set, but it needs to be stated explicitly.


7. The 3D scan imagery has been converted into false-colour images. It is not made clear whether this was a lossy or lossless process. It is not clear where the raw 3D scan data can be obtained, and whether it is permissible to make submissions to the SGxP leaderboards using different preprocessing of the 3D scan data than the false-colour images used by the authors, or a model that takes a 3D point-cloud data structure directly as its input.

**Relation To Prior Work:**

I think this is adequate.

**Summary And Contributions:**

The paper presents a new dataset pertaining to the sorghum genus of agricultural crops. The input modality is visual, consisting of both visible-light RGB photographs and false-color representations of 3D laser-scans of various types of sorghum crops across the lifespan of the plants. The target outputs are the state of several thousand single nucleotide polymorphism (SNP) sites on the DNA of the plant which was imaged. The benchmark is structured such that each SNP is considered its own task, and hence the dataset is comprised of 2732 separate binary tasks. A small number (15) of the SNPs are well-understood by biologists and hence the phenotype that this genotypic change corresponds to is already known, however the phenotypic outcomes of most SNPs in the dataset are not yet understood. The authors hope this dataset will help increase understanding of the phenotypic changes manifested by these genotypes.

**Edit**: The authors have address many of the concerns I raised, and I have raised my score accordingly.

---

> ### Author Response · Authors · 2023-08-21
>
> We thank reviewer VdiJ for their incredibly thorough and thoughtful review. In the interest of space, we have rephrased the reviewers questions/concerns and listed our responses below:
>
> _Q: Are there multiple SNPs at the same location?_
> Every task in the dataset corresponds to a specific variation at a SNP – for example, the SNP sobic_001G269200_1_51588525 is a SNP on Chromosome 1 at position 51588525 where the reference is a T and the alternate is a C.
>
> _Q: Can you better describe the distribution of samples per SNP?_
> We have replaced Table 2 with a new Figure showing histograms of images per label for the training and reference datasets, as well as a histogram of the number of reference minus the number of alternate examples, to better capture the class imbalance.
>
> _Q: Why are two output logits used?_
> Using two logits lends itself to simpler class visualization approaches, which are useful in discussing what the models appeared to focus on with biologists. We did not observe significant overfitting, indicating no issues with overparameterization.
>
> _Q: Why don’t you show performance with fine-tuning?_
> We finetune a small layer on top of the frozen encoder, but agree that fully fine-tuned task-specific models are likely to outperform that. Our goal, however, is not to achieve the best possible performance on all possible tasks, but rather to provide a reasonable point of comparison across all tasks for anyone using this benchmark.
>
> _Q: Pre-processing and availability of the 3D data?_
> We reference the DataDryad link for the TERRA-REF data both in the full paper and in the Datasheets for Datasets documentation in the Appendix which has info on downloading the 3D data.  We have updated the paper to clarify that submissions to the leaderboards may use different preprocessing, and that it should be documented in the required code as part of submission to the leaderboard.
>
> _Q: Some SNP’s have no phenotypic impact?_
> We have updated our paper to highlight this limitation, which is a fundamental challenge in GxP exploration.  The relevant computational problem is to find candidate SNPs that are compelling, and additional (biological) work is necessary to tease out the full GxP interactions.
>
> _Q: Determining phenotypic impact is non-trivial?_
> The difficulty of determining what phenotypic features it detects _is_ non-trivial – that is why the sorghumsnpbenchmark website contains discussion boards for every SNP in addition to the leaderboard.  In lieu of mandating a particular approach, we offer a generic discussion board to allow participants to share what they find in whatever way they can.  We have updated the limitations section to reflect more on this.
>
> _Q: Non-visible phenotype changes?_
> We have added an acknowledgement that this benchmark is limited to above ground traits that are visible in RGB and 3D data.
>
> _Q: Unclear on utility to general ML community?_
> Researchers in the machine learning community that want a “standard” benchmark should focus on the 15 SNPs with known phenotypic expression.  We acknowledge that the exploratory nature of the SNPs with unknown effect makes it more difficult to use those results as a contest. The popularity of science focussed workshops (such as ML4PhysicalSciences and AI for Science Workshops at NeurIPS, the Agriculture-Vision and EarthVision Workshops at CVPR, the Computer Vision Problems in Plant Phenotyping and Agriculture Workshop at ICCV and ECCV, the ICML Workshop on Computational Biology) suggest that ML researchers have broad interests.
>
> _Q: Repeated RGB images in the multi-modal data?_
> We have included further details on the multimodal construction in the appendix. We maximized the number of multimodal image pairs included in the dataset (removing only those where the 3D scanner image did not have another image from the same plot within 10 days), while allowing for more repeated RGB images (when there were fewer RGB images captured in the same plot within 10 days).
>
> _Q: Why isn’t the data setup to train on all SNPs at once?_
> Different choices had different trade-offs, we optimized for the inclusion of more SNP’s to maximize potential genotype x phenotype discovery.  The additional dataset from a different season allows approaches that train a single model.
>
> _Q: Train/test splits?_
> The protocol for performing dataset splits is described in detail in the Appendix.
>
> _Q: The age of the plant/crop life-cycle might be important._
> We include timestamp information in the metadata for each image, allowing participants to choose if they would like to incorporate temporal information into their model.
>
> _Q: Is there information leakage in the pretraining approach the baseline model uses?_
> The pretraining data comes from an entirely different season with entirely different cultivars, so there is no risk of SNP label leakage.
>
> _Q: Typos/citation issues:_
> We appreciate the reviewer identifying typos and issues in our citations – we have addressed these in the updated document.

---

> > ### Comment · Reviewer_VdiJ · 2023-08-25
> > **General response to authors (presentation of work)**
> >
> > Thanks to the authors for taking my feedback onboard, updating their manuscript, and responding to several of the points I raised. It seems there have been quite a lot of changes, but it is tricky to tell which bits of the paper have been changed. Could the authors upload a copy of the PDF with new/modified text highlighted in a different colour to make it easier to find changes?
> >
> > I will provide some responses here, request follow-up on points that were not addressed, and request new follow-up where appropriate. Due to the [character count of my responses](https://www.quora.com/Who-wrote-the-quote-If-I-had-more-time-I-would-have-written-you-a-shorter-letter) I will de-thread the discussion into a series of comments on particular (broad) topics which can form the basis of new comment threads.
> >
> > > *Q: Are there multiple SNPs at the same location?* Every task in the dataset corresponds to a specific variation at a SNP – for example, the SNP sobic_001G269200_1_51588525 is a SNP on Chromosome 1 at position 51588525 where the reference is a T and the alternate is a C.
> >
> > Is this now explained in the paper? If so, can you point me to the line number? e.g. The total number of SNP locations and number of SNP variations in the dataset.
> >
> > > *Q: Can you better describe the distribution of samples per SNP?* We have replaced Table 2 with a new Figure showing histograms of images per label for the training and reference datasets, as well as a histogram of the number of reference minus the number of alternate examples, to better capture the class imbalance.
> >
> > Thank you for adding this Figure 3. This is much more informative than the table was! However, please increase the font size of the axes ticks, axes labels, and title text. Each should be no smaller than 80% of the size of the text in the main body of the paper. Change the x-axis units to be in thousands so it is more legible too. It is easier to read "300 [axis label: Number of images (thousands)]" than "300000 [axis label: # images]". I shouldn't have to count the number of 0s in the axis ticks to understand the data in the figure - I can't tell the difference between 4-6 zeros in a row at a glance and I'm sure you can't either. The figures would also be easier to read with major/minor y-grid lines enabled. For Figure 3c, should this be a shown as a relative number instead of the absolute difference? A difference of 20k images would look small on the current scale, but represents a large imbalance if there's only 50k images to use for that SNP, so I think relative difference (histogram of what % of train data is alt. class) would be more informative. A 2d histogram would really show the data distribution completely (# reference samples vs # alt samples on x and y with colour acting as the z-axis for the bins to show frequency) and would be ideal(?), but is not necessary (don't feel obligated to implement a 2d histogram, but go ahead if you want to).
> >
> > > *Q: Some SNP’s have no phenotypic impact?* We have updated our paper to highlight this limitation, which is a fundamental challenge in GxP exploration. The relevant computational problem is to find candidate SNPs that are compelling, and additional (biological) work is necessary to tease out the full GxP interactions.
> >
> > Where do I find this? (What line number?)
> >
> > > *Q: Non-visible phenotype changes?* We have added an acknowledgement that this benchmark is limited to above ground traits that are visible in RGB and 3D data.
> >
> > Line number?
> >
> > > *Q: Why isn’t the data setup to train on all SNPs at once?* Different choices had different trade-offs, we optimized for the inclusion of more SNP’s to maximize potential genotype x phenotype discovery. The additional dataset from a different season allows approaches that train a single model.
> >
> > Please mention in the paper?
> >
> > > *Q: Is there information leakage in the pretraining approach the baseline model uses?* The pretraining data comes from an entirely different season with entirely different cultivars, so there is no risk of SNP label leakage.
> >
> > I see this is clarified in the paper, thank you.
> >
> > X1: I still think it would be a good idea to mention that sorghum is a genus of grass plants, and that its food uses include both human and livestock consumed foods.
> >
> > X2: Please add a description of each of the columns in your metadata CSV files somewhere. Either on the dataset landing page, or in a README file within the metadata tarball, or both. It is not clear what the significance of original_id, subpopulation, cultivar, and plot are. It is not clear what the axes and units are for camera_x, camera_y, camera_z. It is not clear what the timezone is for timestamp. Is it the local time? Will there be a discontinuity with local daylight savings time changes? If it is not the local time, I will need to know the timezone.

---

> > > ### Author Response · Authors · 2023-08-28
> > >
> > > We thank the reviewer for their continued engagement and respond in line below.
> > >
> > > >Q: Are there multiple SNPs at the same location? Has this been updated in the text? Total number of SNP locations and number of SNP variations in the dataset?
> > >
> > > A: We have updated the first paragraph in Section 3 (lines 148-159) to be explicit about this. Because there are no two SNPs at the same location, the total number of SNP locations in the benchmark is equal to the total number of SNPs in the benchmark (the 2,717 SNPs referenced throughout the paper).
> > >
> > > >Q: Improvements to Figure 3.
> > >
> > > A: We appreciate the suggestions on how to improve Figure 3 and have updated the figure in a way to clarify the distributions of samples per SNP. We have modified the font sizes and made Figure 3 (c) normalized by the total number of images in the dataset. We also have updated the text in the paragraph at line 219 to discuss this figure.
> > >
> > > >Q: Some SNP’s have no phenotypic impact? We have updated our paper to highlight this limitation, which is a fundamental challenge in GxP exploration. The relevant computational problem is to find candidate SNPs that are compelling, and additional (biological) work is necessary to tease out the full GxP interactions. Where do I find this? (What line number?)
> > >
> > > A: This is in Section 7: Limitation, lines 361-365
> > >
> > > >Q: Non-visible phenotype changes? We have added an acknowledgement that this benchmark is limited to above ground traits that are visible in RGB and 3D data. Line number?
> > >
> > > A: This is in Section 7: Limitation, lines 357-359
> > >
> > > >Q: Why isn’t the data setup to train on all SNPs at once? Different choices had different trade-offs, we optimized for the inclusion of more SNP’s to maximize potential genotype x phenotype discovery. The additional dataset from a different season allows approaches that train a single model. Please mention in the paper?
> > >
> > > A: This is in Section 4, Baseline Approach:  lines 257-262
> > >
> > > >X1: I still think it would be a good idea to mention that sorghum is a genus of grass plants, and that its food uses include both human and livestock consumed foods.
> > >
> > > We appreciate this and updated in the first sentence of 2.3 accordingly.
> > >
> > > >X2:  Please add a description of each of the columns in your metadata CSV files somewhere. Either on the dataset landing page, or in a README file within the metadata tarball, or both
> > >
> > > We have added this information to the appendix, as well as in a metadata readme file, which includes definitions for subpopulation, cultivar and plot, and information such as the data type and timezone. It can be accessed from the sorghumsnpbenchmark website (alongside the metadata files) or directly at https://cs.slu.edu/~astylianou/neurips_sorghum_dataset/metadata_readme.txt.

---

> > ### Comment · Reviewer_VdiJ · 2023-08-26
> > **Partitioning data**
> >
> > > *Q: Are there multiple SNPs at the same location?* Every task in the dataset corresponds to a specific variation at a SNP – for example, the SNP sobic_001G269200_1_51588525 is a SNP on Chromosome 1 at position 51588525 where the reference is a T and the alternate is a C.
> >
> > I understand there are some trade-offs which prevent you from using a single train/test split. Can I request that the reference images used for SNP variants at the same location do not collide? i.e. The test images for one variant should not be in the train set for another variant at the same SNP location. This will mean that a researcher can train a multi-class model on that SNP location and be compatible with submitting to the benchmark.
> >
> > > *Q: Repeated RGB images in the multi-modal data?* We have included further details on the multimodal construction in the appendix. We maximized the number of multimodal image pairs included in the dataset (removing only those where the 3D scanner image did not have another image from the same plot within 10 days), while allowing for more repeated RGB images (when there were fewer RGB images captured in the same plot within 10 days).
> >
> > Thank you for adding the details for the multimodal dataset construction. Has the methodology changed at all? I still do not think it is appropriate to repeat some RGB images so 100s of times more than others, and your methodology needs to be changed. Why do we want the 3d images completely balanced yet RGBs so very imbalanced? The multimodal test set really needs to be better constructed. I say it should contain only a single instance of each RGB image and each 3D image (i.e. an image from a cultivar selected for the test set occurs either 0 or 1 times across the multimodal test set), but balanced could be achieved otherwise. The training set "pairs" are less important.
> >
> > Here is my suggestion:
> > - Use the cultivars selected for the test set as already performed.
> > - For each 3D image, find the K temporally nearest RGB image and store the tuple of all image pairs as (delta_time_in_seconds, image_path_3d, image_path_rgb). An appropriate value for K might be K=1 or K=2, higher values permit more leapfrogging of neighbours.
> > - For each RGB image, find the K temporally nearest 3D image and store the tuple of all image pairs as (delta_time_in_seconds, image_path_3d, image_path_rgb)
> > - Concatenate the two lists together, and sort ascending by delta_time_in_seconds.
> > - Pop the row/tuple with the shortest duration off the top of the stack and add it to the test set. Delete all remaining pairs which featured either of these two images from the stack. Repeat until the stack is empty or delta_time_in_seconds exceeds 10 days.
> >
> > Please tell me if you see a reason why having a balanced multimodal test set would not be viable.
> >
> > For the multimodal training images, providing the exact pairing to use seems overly prescriptive. I think it is fair to say that all images in the RGB and 3D image training sets are fair game and let the user pair the images together as they wish. Training image pairs can be provided (ideally modified to be better balanced between repeating RGB images and never repeating 3D images), but it should be permissible to use other pairings of images in the train set as the user sees fit. Randomly pairing together images within some temporal window should provide a data augmentation boost when training.
> >
> > Whichever way you handle multimodal partitioning, either you need to redact the single-modal RGB and 3D image test sets down to the same set of images used in the mulitmodal test set (not ideal as it means there will be fewer test samples for these partitions), or you need to mention clearly in the paper that the set of images contained in the test partitions is different. Otherwise people may expect things like "performance of multimodal model" > "performance of RGB model" implies 3D contain information not in RGB images, when this might just be an illusory improvement caused by the evaluation set unexpectedly changing.
> >
> > > *Q: Train/test splits?* The protocol for performing dataset splits is described in detail in the Appendix.
> >
> > It's in the "Datasets for Datasheets" form, which I didn't read because I view that as a summary of the dataset facts, not the full details about the dataset construction process. The answer to "Are there recommended data splits (e.g., training, development/validation, testing)" should IMO be one word only ("Yes"), or "Yes, see Section X.Y", not the only source of information about how the splits were constructed - that information should appear elsewhere in the paper. If the full details about the partitioning scheme are only in the Appendix due to space, the main text needs to have some brief description (balanced 70/30 split of cultivars) and direct the reader to the appropriate section of the appendix for more information. I shouldn't read the whole of the main paper and come away from that with no idea how you partitioned the train/test splits.

---

> > > ### Author Response · Authors · 2023-08-28
> > >
> > > >Q: Are there multiple SNPs at the same location? Can I request that the reference images used for SNP variants at the same location do not collide?
> > >
> > > A: We replied in part to this in reply to your other comment/question on this matter. We apologize if our original response was not entirely clear on this. While the reviewer is correct that it is biologically possible to have two SNPs at the same location, this does not occur in our benchmark.  We have updated the first paragraph in Section 3 (lines 148-159) to be explicit about this. Because there are no two SNPs at the same location, the total number of SNP locations in the benchmark is equal to the total number of SNPs in the benchmark (the 2,717 SNPs referenced throughout the paper).
> > >
> > > > Q: Please tell me if you see a reason why having a balanced multimodal test set would not be viable.
> > >
> > > A: As we constructed this dataset, we evaluated a number of design choices.  The design of the multi-modal dataset was one that had a number of trade-offs. We considered these issues extensively when we created the dataset.  Lines 247-259 in the paper share some of our thoughts about this, and we expand on it in Section A.2 of the appendix.   Our criteria for choosing between different options of creating the data was to maximize the amount of data available.
> > >
> > > We implemented almost exactly the strategy that you (the reviewer) suggest, which led us to the following trade-off:
> > >
> > > * \# of 3D scanner images in the multimodal benchmark if we allow repeated RGB images: 528,038
> > >
> > > * \# of 3D scanner images in the multimodal benchmark if we don’t allow RGB images to be repeated: 305,185
> > >
> > > Because this was substantially more 3D scanner images that would be included, and since the zeitgeist of the modern ML era is that more data is better, we decided on the approach in our paper.  We recognize that repeating RGB images in the dataset creates its own set of challenges; most RGB images appear only once, although many RGB images appear a few times, and a few appear many times (the median is 1.0 appearances for an RGB image, the mean is 1.52 with a standard deviation of 1.78).
> > >
> > > We felt that giving specific pairs of RGB and 3D imagery made the dataset approachable to a bigger share of the ML community, and within the constraints of approaches to constructing these image pairs, the trade-off of more imagery with some re-use was the better trade-off.
> > >
> > > >Q: Why is your dataset splitting process only in the Datasets for Datasheets? In the Datasets for Datasheets, it should just be a yes/no answer.
> > >
> > > A: We have added the training/testing split explanation to Section 4. Regarding the inclusion of the full description of the splitting procedure in the Datasets for Datasheets documentation, we will note that the original Datasets for Datasheets paper specifically requests that the full description should be included in the documentation:
> > >
> > > “**Are there recommended data splits (e.g., training, development/validation, testing)? If so, please provide a description of these splits, explaining the rationale behind them.**”
> > >
> > > But nonetheless, we appreciate that it’s helpful to have the splitting information in the text directly and not just the appendix, and so have added it. It can be found on lines 219-226.

---

> > > > ### Comment · Reviewer_VdiJ · 2023-08-30
> > > >
> > > > What is/are the scientific question(s) you are hoping to answer with the multimodal test set? I have been assuming that you want to achieve the following:
> > > >
> > > > - ascertain whether there is visually discriminable information about an SNP (test model performance >50%, by more than chance level)
> > > > - distinguish whether one model contains more information about the visual characteristics of an SNP than another model (test model performance > other model performance, by more than chance level)
> > > > - establish whether there is information about an SNP in the RGB imagery which is not contained in the 3D structure imagery (model performance on multimodal > model performance on 3D images, by more than chance level)
> > > > - establish whether there is information about an SNP in the 3D imagery which is not contained in the RGB imagery (model performance on multimodal > model performance on RGB images, by more than chance level)
> > > >
> > > > If I am wrong about this, please correct me and you need to add a very clear disclaimer which says the multimodal test set is not intended to be compared with single-modal test performances, since nobody would assume that.
> > > >
> > > > The way I see it, your current test set is deficient at answering whether there is information in the RGBs that isn't in the 3D imagery because the 3D images because some 3D images only appear in one test set, and much more deficient at answering whether there is information in the 3D imagery that isn't in the RGB images because some RGB images appear only in one test set and others appear multiple times.
> > > >
> > > > I appreciate that there are trade-offs creating the multimodal dataset given that the images weren't paired at the time of acquisition. However, I still think the multimodal test set is not acceptable in its current state.
> > > >
> > > > > and since the zeitgeist of the modern ML era is that more data is better,
> > > >
> > > > That's more *training* data, to support building bigger, better, more generalizable models - not more test data. I already said I think the multimodal train and test sets can be created in different ways. (You can share a pre-mixed training set that has repeated images without sharing a test set that has repeated images.)
> > > >
> > > > In general, for the test set it is important that it can ask a question that you are looking to answer. Usually we care about the ability of the model to generalize to unseen samples from within the same domain as the training set. In your case, you've identified that you want to see whether the model can generalize to unseen cultivars that do vs do not possess the mutation at that SNP. The size of the test set needs to be big enough to evaluate the performance of the model to a sufficiently high level of precision - for example to evaluate two candidate models, compare their performances, and ascertain whether one model is better at the task that the other. Bigger is is better for test sets in that you can make stronger statements about which model performs best on the test set if it is larger. But that's of no use if there is systematic bias in the test set which prevents it from addressing the question it is there to ask. Making the test set bigger without taking care to note what is added may be unwise. Obviously, if you make it bigger by adding pseudo-replicas of samples in the training set to the test set (adding a mirror image of every image in the training set to the test set, say), that's unwise as it is going to hamper your ability to estimate the generalization performance of the model. If you add more samples to the test set but all taken from one part of the sample domain, even if it is orthogonal to the classification task (e.g. you add a lot of samples of white male faces to your dataset that's supposed to predict the age of faces of all ethnicities), that's going to screw up your test evaluation too. This is the type of problem you are imposing on your own test set right now. The RGB images selected to go into the MM test set were not selected with uniform weighting. There is a systematic bias that means that images that are at the start or end of a recording sequence are over-represented. I'm not sure what that means in practice, but probably that images at the edge of the plot, and at the start and end of the day are more predominant. That might result in the lighting being different, or the plant's behaviour may change at the start or end of the day (more open/closed). These factors may effect the RGB model's performance in general because less of the plant is visible in the image. Whether this is how the problem will manifest, I do not know, but we do know that the test set has been inflated by adding replicated images to the test set that are sampled in a systematic way, thus imposing a systematic bias upon it.

---

> > > > > ### Comment · Reviewer_VdiJ · 2023-08-30
> > > > >
> > > > > Your current text in the paper says that this is only one way of pairing the data and encourage practitioners to explore other pairings, and I think randomly pairing together images during training would be a sensible data augmentation strategy and something people will explore if they are working with this dataset (especially if you explicitly say this is permitted). But realistically, people aren't going to make new test sets on their own when you've already established a test set for the dataset, shared it as part of the dataset download, and created an online leaderboard track for it. Users of the dataset aren't going to feel empowered to change the test set from the official test set, and if they do their test set construction is unlikely to be widely distributed. Hence there is onus on you to get the test set right. You could fix the test set later it by releasing a v2 of the dataset, but it would be better to fix the problems now instead when releasing v1.
> > > > >
> > > > > You could fix most of the problems by redacting the existing MM test set down to only have one appearance of each RGB image, or at most two appearances of each RGB image. Though it is not the best solution, this resolves the most egregious problems and does so straight-forwardly.
> > > > >
> > > > > I am reluctant to increase my rating of the paper while the biggest problem I had with the dataset remains unchanged.

---

> > > > > > ### Author Response · Authors · 2023-08-30
> > > > > >
> > > > > > The scientific question that we are attempting to explore with the multimodal dataset is not whether one modality is better or worse for a specific SNP necessarily – rather we believe that there may be complex phenotypes that are observable using multiple modalities that may not be apparent using only one modality. The relative importance of different modalities certainly may be an observation from the multimodal analysis, but these more complex phenotypes are the reason we are most excited to include a multimodal benchmark. We have added text clarifying this motivation on lines 210-214.
> > > > > >
> > > > > > We appreciate the reviewer’s suggestion to treat the multimodal training and testing datasets separately. As such, we are in the process of updating our multi-modality testing datasets to eliminate any repeated RGB images and computing updated test accuracies using these datasets. We will still allow repeated RGB images in the training datasets to maximize the number of image pairs available during training. We have updated the appendix to document this procedure, and include it here for convenience:
> > > > > >
> > > > > > > 1. For every RGB image, initialize a match_counter to 0 (the number of times this RGB image was used as a match for a 3D scanner image).
> > > > > > > 2. For every 3D scanner image:
> > > > > > >>     a. Find the RGB images (from the respective single modality dataset) that come from the same plot, were captured within 10 days of the 3D scanner image.
> > > > > > >>     b. Remove from this subset any images whose match_counter is greater than the minimum match_counter value for the entire subset.
> > > > > > >>     c. Select one of the remaining images as a good match.
> > > > > > >>     d. Increment the match_counter for that image.
> > > > > >
> > > > > > >Some 3D scanner images have no valid RGB image from the same plot within 10 days. These images are excluded from any multimodal datasets. This construction maximizes the number of multimodal image pairs that are included in the dataset (removing only those where the 3D scanner image did not have another image from the same plot within 10 days), while allowing for more possible repeated RGB images (when there were fewer RGB images captured in the same plot within 10 days).
> > > > > >
> > > > > > >After creating the list of (3D scanner, RGB) pairs using the above methodology, we create SNP-specific multimodal datasets. To do this, we load the 3D scanner test set for a specific SNP. For every image in that test set, we look up its (3D scanner, RGB) pair. If a valid pair exists, then it is added to the possible multimodal test set for that SNP.
> > > > > >
> > > > > > >If the dataset is a testing dataset, we then identify any (3D scanner, RGB) pairs where the RGB image is repeated. We select one of the pairs at random to keep, and remove any other pairs, so that the test set does not contain any repeated RGB images. We allow repeated RGB images in the training set to maximize the number of (3D scanner, RGB) pairs available at training time. We finally rebalance the test set using the same procedure as the single modality datasets, guaranteeing that there are an equal number of reference and alternate examples.
> > > > > >
> > > > > > >Researchers are welcome and encouraged to explore different constructions of multimodal datasets which can be evaluated using the per-cultivar evaluation metric.
> > > > > >
> > > > > > We do want to clarify one of the reviewer’s concerns about our sampling. The reviewer states:
> > > > > > >“There is a systematic bias that means that images that are at the start or end of a recording sequence are over-represented.”
> > > > > >
> > > > > > When we select an RGB image as a match for a 3D scanner image, we always select *at random* an RGB image that (a) is from within 10 days and from the same plot and (b) is least represented thus far when constructing the dataset. We do not always pick the first or last image from some sequence.
> > > > > >
> > > > > > We have implemented the above strategy for avoiding repeated RGB images in the test sets. We are in the process of running it for all of the SNPs in the benchmark, and computing updated accuracies. While we are unlikely to have those numbers by the end of the discussion period today, we will update them in the paper assuming it is accepted, and will update both the dataset files and accuracies on the benchmark website as soon as we have them.
> > > > > >
> > > > > > We genuinely appreciate the reviewer’s continued engagement with us on this matter, and believe the benchmark is stronger for their input.

---

> > > > > > > ### Comment · Reviewer_VdiJ · 2023-08-31
> > > > > > >
> > > > > > > > When we select an RGB image as a match for a 3D scanner image, we always select at random an RGB image that (a) is from within 10 days and from the same plot and (b) is least represented thus far when constructing the dataset. We do not always pick the first or last image from some sequence.
> > > > > > >
> > > > > > > I may have missed this in the revised manuscript. This does mitigate some of my concerns about the systematic bias in the sampling. 20 days (+/-10 days) seems like a rather large window to select from with a uniform distribution to me, though I am not an expert in growing crops. In this case, I think the test set could be constructed better still by prioritizing smaller pairing distances over long ones, and I encourage the authors to refine the test set further in that direction.
> > > > > > >
> > > > > > > Given the sampling method used, how do RGB images with 100s of appearances arise? There must be long windows of time where the ratio of 3D to RGB images becomes very high, which can only reasonably be explained by RGB images becoming very sparse. This suggests a different way to try to select the test set may be... to address this by stratifying across time? e.g. Draw up some temporal bin edges and sample N pairs within each temporal bin, minimizing repetition of images as much as possible but ensuring there are N pairs from each temporal bin even if some samples have to be repeated. This test set places a different emphasis on what the scientific question is, of course, and if the unimodal test sets are not stratified, it is probably not desirable to stratify the multimodal test set by age as it would impact the ability to compare performance on the two test sets.
> > > > > > >
> > > > > > > Anyway, thanks to the authors for addressing many of my concerns, either by rebuttal or modification to the paper. I have raised my review score accordingly.

---

> > ### Comment · Reviewer_VdiJ · 2023-08-28
> > **Crop life-cycle information**
> >
> > > Q: The age of the plant/crop life-cycle might be important. We include timestamp information in the metadata for each image, allowing participants to choose if they would like to incorporate temporal information into their model.
> >
> > Yes, I am aware the metadata is provided. Participants are allowed to incorporate temporal information into their model? That was not the impression I got from reading the paper, nor from the website. Can I train a model that takes the age of the plant as an input as well as the image, $p(\text{class}|\text{image},\text{age})$ instead of $p(\text{class}|\text{image})$ and submit the performance of that model to the benchmark? If so, this needs to be explicitly pointed out!
> >
> > I think there would be an advantage to being able to break down performance by crop age in the future, even if this is not imminently possible. With that in mind, I think it would be helpful for benchmark submissions to have the prediction for each test sample so this can be stored for posterity and analyzed later. Do you think this would be feasible?

---

> > > ### Author Response · Authors · 2023-08-28
> > >
> > > On lines 315-319 (in the updated manuscript, although this text has been in the paper all along), we explain that one of the primary reasons for a per-cultivar evaluation is to give researchers flexibility in exactly what information they use when computing their accuracy:
> > >
> > > > They could, for example, only consider images from a particular portion of the growing season, or weight images from different parts of the growing season differently. They could develop model architectures that consider multiple images simultaneously. In general, the per-cultivar evaluation provides researchers with the ability to develop models that explain complex genotype x phenotype relationships in sorghum.
> > >
> > > We appreciate that we could be even more explicit in stating that participants can use the provided metadata, and so have updated the text (lines 210-218):
> > >
> > > >For every image including in any dataset in the SGxP Benchmark, we also provide additional metadata: the sorghum cultivar and subpopulation shown in the image, the TERRA-REF plot from which the image was captured, and the timestamp that the image was captured. Additional information about this metadata can be found in the Appendix. Benchmark participants are encouraged to explore ways to incorporate this metadata information in their approaches.
> > >
> > > We have also added the metadata description (detailing the definition of cultivar, subpopulation, plot and timestamp) to the Appendix (in addition to linking to it as a readme from the sorghumsnpbenchmark website and github repository).
> > >
> > > We will consider options to solicit predictions per sample in the future, as we agree that this likely will include interesting information about a model's performance, but we currently do not evaluate per-image submissions on the benchmark server (we discussed the reasons for this choice in our overall reply to all reviewers and replies to individual reviewers who mentioned that reviewers can submit their own scores). We anticipate that information like "this model/feature seems very responsive to growth phase/time of season" would be an interesting topic of conversation in the discussion for a particular SNP!

---

> > > > ### Comment · Reviewer_VdiJ · 2023-08-30
> > > >
> > > > > On lines 315-319 (in the updated manuscript, although this text has been in the paper all along), we explain that one of the primary reasons for a per-cultivar evaluation is to give researchers flexibility in exactly what information they use when computing their accuracy
> > > >
> > > > So, you can't use a model conditioned on the age of the plant to submit to the per-image benchmark...? Only for submitting to the per-cultivar track?

---

> > > > > ### Author Response · Authors · 2023-08-30
> > > > >
> > > > > > So, you can't use a model conditioned on the age of the plant to submit to the per-image benchmark...? Only for submitting to the per-cultivar track?
> > > > >
> > > > > We agree with the reviewer that the *previous* text only implied being able to use the external metadata for the per-cultivar task but believe the updated text on lines 210-218, which introduces the entire benchmark and all of its its tasks, is clear that there is no such limitation:
> > > > >
> > > > > > For every image including in any dataset in the SGxP Benchmark, we also provide additional metadata: the sorghum cultivar and subpopulation shown in the image, the TERRA-REF plot from which the image was captured, and the timestamp that the image was captured. Additional information about this metadata can be found in the Appendix. Benchmark participants are encouraged to explore ways to incorporate this metadata information in their approaches.

---

### Author Response · Authors · 2023-08-21
**Thank you to our reviewers.**

We sincerely thank all of our reviewers for their significant effort in providing us with their reviews – these were some of the most thoughtful and thorough reviews that any of the authors have received at any machine learning conference.

We are excited that all of the reviewers see the benefits of this benchmark for genotype x phenotype discovery, and that this is an exciting opportunity for the machine learning community to engage with plant biologists.

We also appreciate the chance to respond to some of the concerns that various reviewers have. In this post, we will respond to high level items that were raised by multiple reviewers. We are also replying in line to each reviewer separately, going into more details on their specific review.


- _Formatting:_ We apologize that we did not include line numbers in the template. Since the benchmark track is not generally anonymized we used the final copy NeurIPS template and did not catch that this disabled things like line numbers. We re-enabled this in our updated version.


- _Dataset Links:_ In an oversight, the sorghumsnpbenchmark.com website contained a reference to the metadata for all of the images used in the benchmark, but was missing the link to the actual images. We have added that link and restructured the webpage to be more obvious what files correspond to images, versus metadata, versus pretraining data, versus pretrained models. We have also updated the github repository to have significantly more documentation and instructions.

    We also appreciated the suggestion from reviewers to include an example dataset so that researchers and practitioners could explore the benchmark without having to download hundreds of gigabytes of files. As such, we have added an example dataset that contains images, metadata, labels and code for a subset of the data (1000 RGB images, 1000 3D images, and 3 markers) so that participants can explore the benchmark without having to download the entire dataset.


- _Allowing participants to self-report accuracy:_ Multiple reviewers expressed concern that reviewers are able to self-report their performance. We debated this at length prior to creating the benchmark. One alternative we considered was having users submit their predictions and computing accuracy on the server. This would be very feasible for per-image accuracy calculations. We, however, want to give as much flexibility in how users determine per-cultivar accuracy as possible (giving flexibility to things like how participants incorporate temporal information or different modalities). This makes computing per-cultivar accuracy on the server infeasible.

    We require participants to provide a link to a codebase in their submission, and our administrators will validate that high performing submissions do not have empty repositories (otherwise they will be removed from the leaderboard). We have updated the submission form to make this requirement, and the consequence of not including a functional repository (removal from the leaderboard) more obvious to participants.

    There’s also marginal benefit to a participant to simply have a high score on the benchmark, limiting the benefit of “spamming” the benchmark with submission. The significance of this benchmark is what comes after achieving that accuracy – interacting with biologists on the sorghumsnpbenchmark forums to actually illuminate what features were important in achieving that accuracy.

   Based on all of these factors, we opted to follow the example of paperswithcode and allow users to self-report performance assuming supporting code is provided.

- _Will the ML community actually use it?_ While we recognize that many ML researchers focus on evaluating general model performance and achieving the maximum accuracy, there are also large and active populations of machine learning researchers that are excited to engage in ML for science, and to explore new machine learning models and explainable AI approaches that can be used to further scientific discovery. Our evidence for this includes not only the significant participation in our prior agriculture-focused competitions and benchmarks, but also the significantly participation in ML for Science venues in recent years,  including the popular ML4PhysicalSciences and AI for Science Workshops at NeurIPS, the Agriculture-Vision and EarthVision Workshops at CVPR, the Computer Vision Problems in Plant Phenotyping and Agriculture Workshop at ICCV and ECCV, the ICML Workshop on Computational Biology, and many, many others. We believe this benchmark will not only be interesting to these communities, but also can be used to learn, understand and develop the best ways that the ML community can work with domain scientists!

We recognize that part of our benchmark is non-traditional and exploratory in nature, and we are excited to share that with the research community. We appreciate the chance to interact with reviewers and improve our submission.

---

### Decision · Program_Chairs · 2023-09-22

**Decision:**

Accept (Poster)

**Comment:**

This paper introduces SG × P (Sorghum Genotype) × The Phenotype Prediction Dataset and Benchmark benchmark provides a large-scale field trial dataset, including large scale high-precision hyper-spectral images and corresponding genomic data. Researchers can use this benchmark test to evaluate phenotype prediction methods based on deep learning. The paper also introduces the organizational structure, evaluation indicators, and performance baseline of benchmark testing.